# Mimicry can drive convergence in structural and light transmission features of transparent wings in Lepidoptera

Charline Sophie Pinna[1]*, Maëlle Vilbert[2], Stephan Borensztajn[3], Willy Daney de Marcillac[4], Florence Piron-Prunier[1], Aaron Pomerantz[5,6], Nipam H Patel[5], Serge Berthier[4], Christine Andraud[2], Doris Gomez[7†], Marianne Elias[1†]

[1]Institut de Systématique, Evolution, Biodiversité (ISYEB), CNRS, Muséum national d'Histoire naturelle, Sorbonne Université, EPHE, Université des Antilles, Paris, France; [2]Centre de Recherche sur la Conservation (CRC), CNRS, MNHN, Ministère de la Culture, Paris, France; [3]Institut de Physique du Globe de Paris (IPGP), Université de Paris, CNRS, Paris, France; [4]Institut des NanoSciences de Paris (INSP), Sorbonne Université, CNRS, Paris, France; [5]Marine Biological Laboratory, Woods Hole, United States; [6]Department Integrative Biology, University of California Berkeley, Berkeley, United States; [7]Centre d'Ecologie Fonctionnelle et Evolutive (CEFE), CNRS, Univ Montpellier, Montpellier, France

**Abstract** Müllerian mimicry is a positive interspecific interaction, whereby co-occurring defended prey species share a common aposematic signal. In Lepidoptera, aposematic species typically harbour conspicuous opaque wing colour patterns with convergent optical properties among co-mimetic species. Surprisingly, some aposematic mimetic species have partially transparent wings, raising the questions of whether optical properties of transparent patches are also convergent, and of how transparency is achieved. Here, we conducted a comparative study of wing optics, micro and nanostructures in neotropical mimetic clearwing Lepidoptera, using spectrophotometry and microscopy imaging. We show that transparency, as perceived by predators, is convergent among co-mimics in some mimicry rings. Underlying micro- and nanostructures are also sometimes convergent despite a large structural diversity. We reveal that while transparency is primarily produced by microstructure modifications, nanostructures largely influence light transmission, potentially enabling additional fine-tuning in transmission properties. This study shows that transparency might not only enable camouflage but can also be part of aposematic signals.

*For correspondence:
ch.pinna@gmail.com

†These authors contributed equally to this work

Competing interest: The authors declare that no competing interests exist.

## Editor's evaluation

This work will likely be of broad interest to evolutionary and ecological researchers, presenting a comprehensive and large-scale comparative study of the evolution of transparency on butterfly wings. The authors find that transparency has repeatedly evolved in mimicry rings, with sometimes similar underlying wing micro and nano structures. The authors suggest that wing transparency may be an aposematic, or warning signal advertising chemical defenses, in addition to camouflage.

## Introduction

Lepidoptera (butterflies and moths) are characterised by large wings typically covered by scales, as testified by the name of the order (after the ancient greek *lepís* - scale and *pterón* – wing). Scales

can contain pigments or generate structural colours, thereby producing colour patterns across the entire wing. Wing colour patterns are involved in thermoregulation (*Dufour et al., 2018*; *Heidrich et al., 2018*), sexual selection (*Kemp, 2007*), and anti-predator defences, such as crypsis (*Cook et al., 2012*; *Endler, 1984*; *Webster et al., 2009*), masquerade (*Skelhorn et al., 2010*; *Stoddard, 2012*), disruptive coloration, and deflection of predator attacks (*Vallin et al., 2011*). Another type of anti-predator defence in Lepidoptera involving wing colour pattern is aposematism, where the presence of secondary defences is advertised by the means of bright and contrasted colour patterns. Because of the positive frequency-dependent selection incurred on aposematic signals (*Greenwood and Cotton, 1989*, *Chouteau et al., 2016*), aposematic species often engage in Müllerian mimetic interactions, whereby species exposed to the same suite of predators converge on the same colour pattern and form mimicry 'rings' (*Müller, 1879*). Co-mimetic species (species that share a common aposematic colour pattern) are often distantly related, implying multiple independent evolution of the same colour pattern. Among such co-mimetic lepidopteran species, several studies using visual modelling have shown that analogous colour patches (i. e. those occupying a similar position in the wing and harbouring similar colour) cannot be discriminated by birds, believed to be the main predators (*Bybee et al., 2012*; *Llaurens et al., 2014*; *Su et al., 2015*; *Thurman and Seymoure, 2016*). Therefore, mimicry selects for convergent (when a trait in different species evolves towards the same value) or advergent (when a trait of a given species evolves towards the trait value in another species) colourations, as perceived by predators.

Surprisingly, although most aposematic Lepidoptera species harbour brightly coloured patterns, some unpalatable (due to the presence of chemical compounds in their body), aposematic species exhibit transparent wing patches (*McClure et al., 2019*). In those species, wing colour pattern typically consists of a mosaic of brightly coloured elements and transparent patches. Notably, in tropical America, many mimicry rings comprise such transparent species (*Beccaloni, 1997*; *Elias et al., 2008*; *Willmott et al., 2017*). Mimicry among species harbouring transparent patches raises the question of selection for convergence in optical properties, as perceived by predators, in those transparent patches.

A related question is whether transparency in co-mimetic species is achieved by the means of similar structural changes in wings and scales. Previous studies on a handful of species (most of which are not aposematic) have revealed several, non-mutually exclusive means to achieve transparency, through scale modification or scale shedding, with the effect of reducing the total coverage of the chitin membrane by scales. Scales can fall upon adult emergence (*Yoshida et al., 1996*); they can have a reduced size (*Dushkina et al., 2017*; *Goodwyn et al., 2009*) and even resemble bristle (*Binetti et al., 2009*; *Hernández-Chavarría et al., 2004*; *Goodwyn et al., 2009*; *Siddique et al., 2015*); they can be either flat on the membrane (*Goodwyn et al., 2009*) or erected (*Dushkina et al., 2017*; *Goodwyn et al., 2009*), which also reduces effective membrane coverage by scales. Reducing scale density could also make wings transparent to some extent (*Goodwyn et al., 2009*). Recently, *Gomez et al., 2021* reported that some Lepidoptera achieve transparency with transparent scales. In addition to scale modifications, the presence of nanostructures on the surface of the wing membrane may enhance transparency through the reduction of light reflection, by generating a gradient of refractive index between the chitin-made membrane and the air allowing better penetration of light through the membrane (*Binetti et al., 2009*; *Siddique et al., 2015*; *Yoshida et al., 1997*). Yet, so far, no study has compared the microstructures (scales) and nanostructures present in transparent patches across co-mimetic species. Furthermore, the diversity of structures described above may lead to a large range of transparency efficiencies. Exploring the link between structural features and optical properties can shed light on whether and how different structures might achieve similar degrees of transparency in the context of mimicry.

Here, we investigate the transmission properties and the structural bases of wing transparency in a community of 62 Neotropical Lepidoptera species belonging to seven families and representing 10 distinct mimicry rings. All mimicry rings contain species with transparent wings, but in a few of them some co-mimics have opaque wings, or nearly so (see *Figure 5—figure supplement 1* for the illustration of this transparency gradient). We characterise wing micro- and nanostructures with digital microscopy and scanning electron microscopy (SEM) imaging and measure transmission properties of transparent patches using spectrophotometry in the range of wavelengths 300–700 nm, visible to both Lepidoptera and their avian predators. We implement comparative analyses that account

for phylogenetic relatedness, to (1) examine the putative convergence or advergence (hereafter, convergence, for the sake of simplicity) among co-mimetic species in visual appearance of transparent patches as seen by bird predators, (2) identify and examine the putative convergence of structures involved in transparency in the different co-mimetic species and finally (3) explore the links between structural features and transmission properties of transparent patches.

## Results

### Convergence among co-mimics in visual appearance of transparent patches as seen by bird predators

To assess whether transparent patches of co-mimetic species were under selection for convergence due to mimicry, we tested whether these transparent patches were more similar, as perceived by predators, among co-mimetic species than expected (1) at random, and (2) given the phylogeny. The first test, which assesses whether predators have a similar perception of analogous transparent patches in co-mimetic species, informs on the selection on transparent patches incurred by predators. The second test, which accounts for the phylogenetic relationship between species, informs on the underlying process leading to similarity, and specifically on whether any case of similarity among co-mimics detected in the first test is due to shared ancestry or to evolutionary convergence. We used spectrophotometry to measure specular transmittance of the transparent patches, which is a

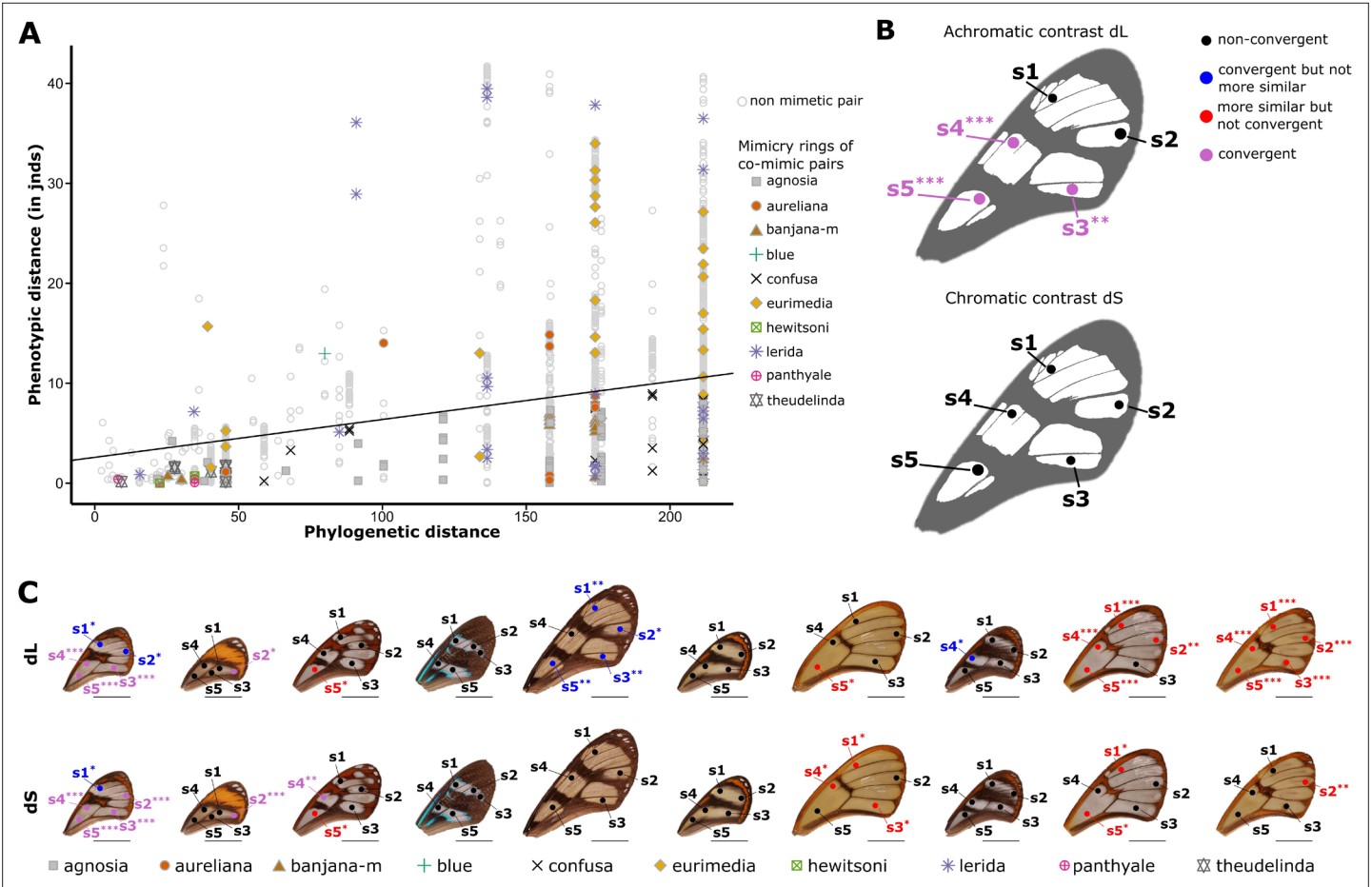

**Figure 1.** Test of convergence of transmission properties between co-mimetic species.

The online version of this article includes the following figure supplement(s) for figure 1:

**Figure supplement 1.** Results of the test of convergence of achromatic (dL) contrasts for each mimicry ring.

**Figure supplement 2.** Results of the test of convergence of chromatic (dS) contrasts for each mimicry ring.

**Figure supplement 3.** Differences in an avian analogue of luminance (quantum catch for brightness channel) between mimicry rings.

quantitative measurement of transparency. As birds are assumed to be the main predators of butter-flies (**Brower, 1984**), we applied bird perceptual vision modelling on the resulting spectra to calculate the chromatic and achromatic contrasts (respectively dS and dL) for each pair of species from our dataset. If transparent patches among co-mimetic species are more similar than expected at random or given the phylogeny, contrasts between pairs of co-mimetic species are expected to be smaller than predicted at random and given the phylogeny, respectively. We only compared analogous spots (i.e. occupying a similar position on the forewing) between species. The results presented in **Figure 1** show that for three spots out of five and across all mimicry rings the difference in achromatic contrast (dL) between co-mimetic species is significantly smaller than expected both at random and given the phylogenetic distances between species (**Figure 1B**), irrespective of the illuminating light or the visual system considered (see **Supplementary file 1a and b** for results under the full range of conditions). Differences in chromatic contrasts (dS) between co-mimetic species are marginally significantly smaller than expected at random and given the phylogenetic distance between species only for the most proximal spot on the forewing (see **Supplementary file 1b**). These results mean that, on average, predators see transparent patches among co-mimetic species as more similar than among species that belong to different mimicry rings. The fact that these tests remain significant (dL) or marginally so (dS) with the phylogenetic correction indicates that such similarity in transparent patches is due to convergent evolution. When looking more precisely at similarity between co-mimetic species for each individual mimicry ring (**Figure 1C**), we show that in six out of 10 mimicry rings achromatic contrasts (dL) between co-mimetic species are smaller than expected at random for at least one spot on the forewing (**Figure 1—figure supplement 1**). After accounting for the phylogeny, this figure drops down to two out of 10 mimicry rings (**Figure 1—figure supplement 1**). Two additional mimicry rings showed evidence for convergence, but not increased similarity. Regarding chromatic contrasts (dS), six mimicry rings out of 10 comprise co-mimetic species exhibiting smaller chromatic contrast than expected at random and three out of them comprise co-mimetic species with smaller chromatic contrast than expected given the phylogeny (**Figure 1C**, **Figure 1—figure supplement 2**). These results suggest that in some cases the similarity in transparent patches between co-mimetic species is due to convergent evolution but we cannot rule out that for some mimicry rings (notably 'theude-linda', 'hewitsoni', 'panthyale') similarity could be due to shared ancestry. A lack of statistical power may also explain why we do not find any convergence or similarity for some mimicry rings as many of these mimicry rings only comprises two (e.g 'blue' mimicry ring) to three (e.g. 'hewitsoni' mimicry ring) species in our dataset. Achromatic aspects (achromatic contrast dL) appear more significant than chromatic aspects (chromatic contrast dS) (**Figure 1**, **Supplementary file 1a,b**), suggesting that selection may act more on broadband transmittance (which is related to the degree of transparency) than on colour in transparent patches.

## Diversity and convergence among co-mimics of structures involved in transparency

Convergence in transmission among co-mimetic species raises the question of the nature and similarity of clearwing microstructures (scales) and nanostructures among co-mimetic species. We therefore explored the diversity of micro- and nanostructures present in the transparent patches in our 62 species. We used digital photonic microscopy and SEM imaging to characterise the structures present in the transparent patches (type, insertion, colour, length, width, and density of scales; type and density of nanostructures; wing membrane thickness).

We found a diversity of microstructural features in transparent patches (**Figure 2A**). Scales could be coloured (76 % of species) or transparent (24%); they could be flat on the membrane (16%) or erected (84%). Scales could be lamellar (55 % of species), or piliform (45%). In our dataset, piliform scales (mainly bifid) appeared to be almost exclusively found in the Ithomiini tribe, although one erebid species also harboured monofid piliform scales (**Figure 3**).

We also revealed an unexpected diversity of the nanostructures that cover the wing membrane (**Figure 2B**). In our sample, we found five types of nanostructures: absent (10 % of species), maze (3%), nipple arrays (55%), pillars (21%), and sponge-like (11%).

Phylogenetic signal tests show that both micro- and nanostructure features are highly conserved in the phylogeny (**Figure 3**, **Supplementary file 1c**), suggesting the existence of constraints in the developmental pathways underlying micro- and nanostructures. However, the value of δ, the metric

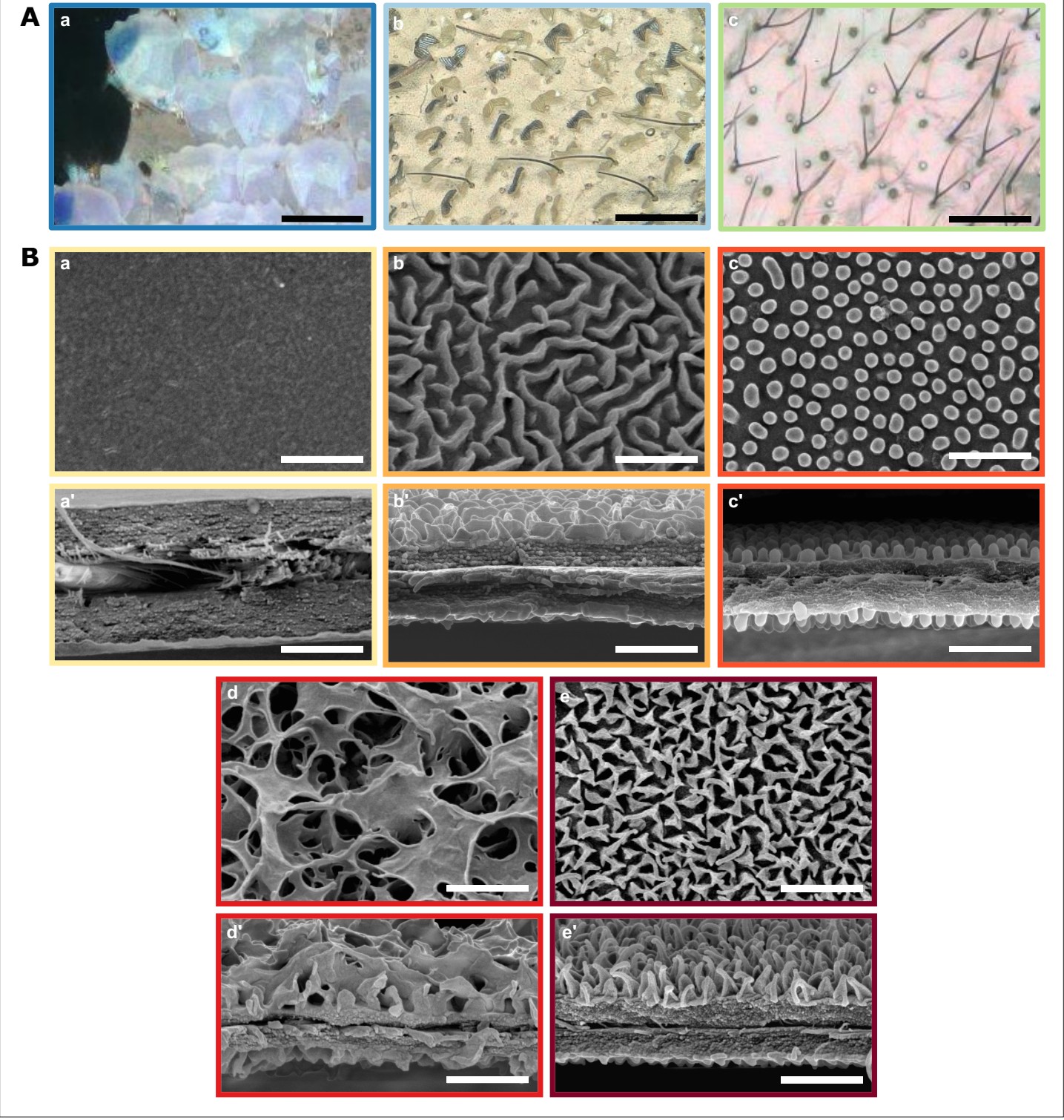

**Figure 2.** Diversity of micro- and nanostructures involved in transparency. (**A**) Diversity of microstructures. (**a**) transparent lamellar scales of *Hypocrita strigifera*, (**b**). erected lamellar scales of *Methona curvifascia* and (**c**). piliform scales of *Hypomenitis ortygia*. Scale bars represent 100 μm. (**B**) Diversity of nanostructures. (**a**), (**b**), (**c**), (**d**) and (**e**) represent top views and (**a'**), (**b'**), (**c'**), (**d'**), and (**e'**) represent cross section of wing membrane. Scale bars represent 1 μm. (**a**), (**a'**). absence of nanostructure in *Methona curvifascia*; (**b**), (**b'**). maze nanostructures of *Megoleria orestilla*; (**c**), (**c'**). nipple nanostructures of *Ithomiola floralis*; (**d**), (**d'**). sponge-like nanostructures of *Oleria onega*; (**e**), (**e'**). pillar nanostructures of *Hypomenitis enigma*. Each coloured frame corresponds to a scale type or nanostructure type, as defined in *Figure 3*.

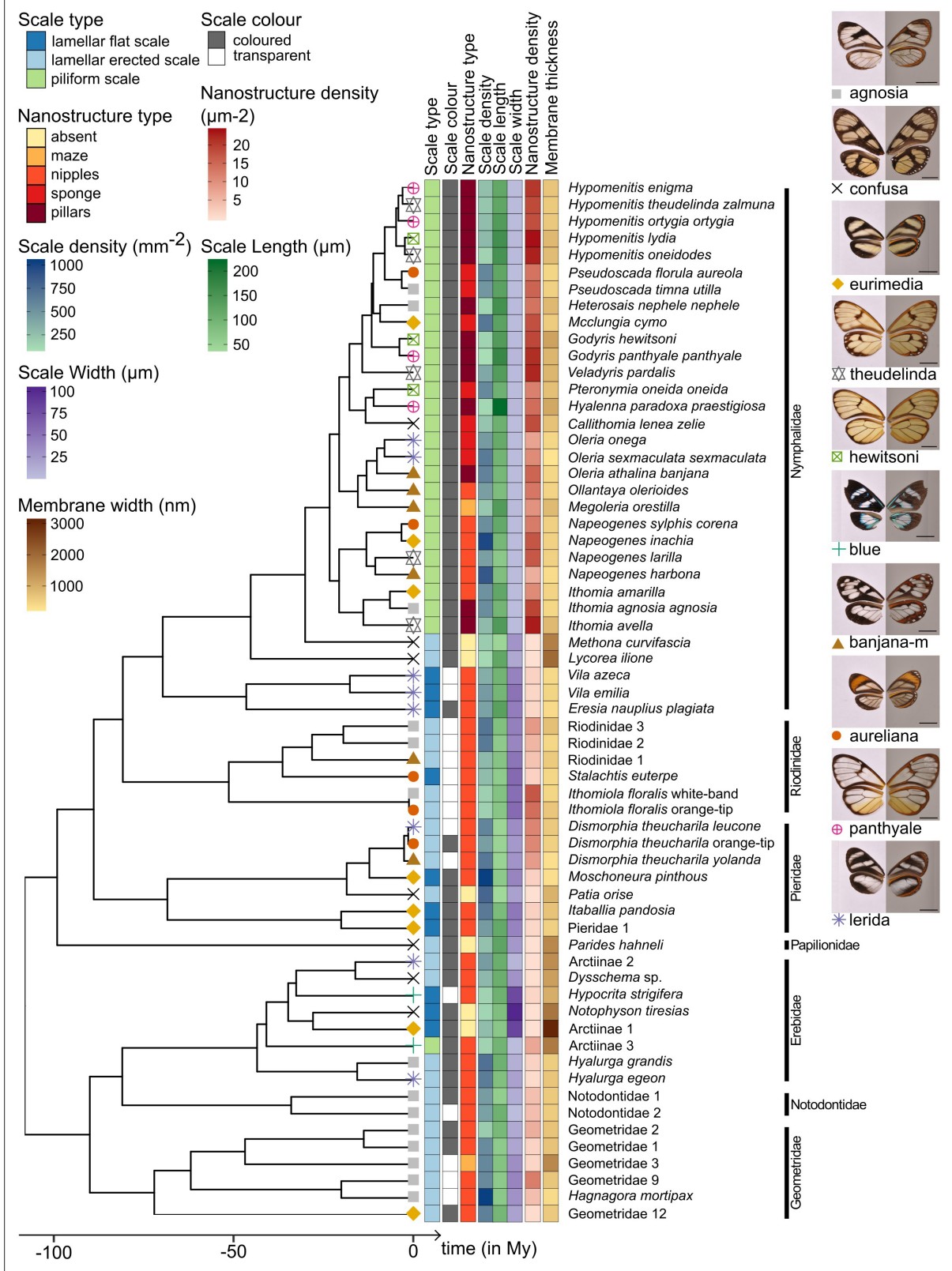

**Figure 3.** Phylogeny of the 62 species considered in this study and distribution of traits along the phylogeny. Mimicry rings are represented by a symbol and a specimen is given as an example for each mimicry ring. Dorsal side of wings has been photographed on a white background (left column) and ventral side on a gray background to highlight the transparent patches (right column). The x axis represents time in million years (My).

*Figure 3 continued on next page*

*Figure 3 continued*

The online version of this article includes the following figure supplement(s) for figure 3:

**Figure supplement 1.** Maximum clade credibility tree of all the specimens obtained with BEAST 1.8.3.

used to quantify phylogenetic signal of traits with discrete states (*Borges et al., 2019*), is higher for scale type than for nanostructures. This means that the phylogenetic signal is stronger for scale type than for nanostructures. Moreover, in the nymphalid tribe Ithomiini, which is highly represented in our dataset, microstructures seem to be more conserved (all species but the basal species *M. curvifascia* have piliform scales in transparent patches) than nanostructures (all five types of nanostructures, mixed in the Ithomiini clade, *Figure 3*).

We then investigated the convergence of structures among co-mimetic species by testing whether co-mimetic species shared structures more often than expected at random and given the phylogeny (see Materials and methods for details). We show that, across all mimicry rings, co-mimetic species share structural features (either scale type, nanostructure type, nanostructure density or *structural syndrome*, defined as the association of scale type and nanostructure type) more than expected at random and given the phylogeny (*Figure 4*). The fact that the tests remain significant when phylogenetic correction is applied means that structural features are globally convergent between co-mimetic species. We tested for convergence of structural features in each individual mimicry ring separately (*Figure 4D* and *Figure 4—figure supplement 1*) and we found that microstructures are convergent for 'agnosia' mimicry ring, where species mainly have erected scales. In other mimicry rings ('panthyale' and 'theudelinda'), species all have similar piliform scales but this similarity is likely due to shared ancestry and not to convergence. Regarding nanostructural type we revealed convergent evolution for 'agnosia', and 'panthyale' mimicry rings, characterised by nipples and by pillars, respectively (*Figure 4D*, *Figure 4—figure supplement 1*). Moreover, we showed that nanostructure density is convergent for 'agnosia', 'confusa', and 'lerida' mimicry rings and that it is more similar than expected at random for 'theudelinda' mimicry ring (*Figure 4D*, *Figure 4—figure supplement 1*). We showed convergence in structural syndrome (association between micro- and nanostructures) for 'agnosia', where 71 % of species harbour a combination of erected scales and nipples (*Figure 4D*, *Figure 4—figure supplement 1*). For the mimicry rings 'panthyale' and 'theudelinda' 100% and 80% of species harbour a combination of piliform scales and pillars, respectively (*Figure 4*), but this similarity is best explained by shared ancestry.

The fact that both transmission properties and underlying structures show some degree of convergence raises the question of whether specific structures have been selected in co-mimetic species because they confer a peculiar visual aspect, typical of the mimicry ring. To address this question, we investigated the link between structural features and transmission properties in transparent patches.

## Link between structural features and transmission properties

To investigate whether transmission properties depend on structural features we used the above measurements of the specular transmittance of transparent patches of each species (see *Figure 5—source data 1* for raw spectra) and we calculated the mean transmittance over 300–700 nm, hereafter called mean transmittance, for each spectrum. The physical property 'mean transmittance' (a proxy for the degree of transparency), is correlated to what is predicted to be perceived by predators based on vision modelling, (see Appendix and *Supplementary file 3a–c* for details), as shown in *Gomez et al., 2021*. Across the 62 species, the mean transmittance ranges from 0.0284 % in *Eresia nauplius* to 71.7 % in *Godyris panthyale* (mean: 29.2%, median: 31.6%, *Supplementary file 1d*). We performed Phylogenetic Generalised Least Squares (PGLS) to assess the relationship between mean transmittance and micro- and nanostructural features (type, insertion, colour, length, width, and density of scales; type and density of nanostructures; wing membrane thickness; including some interactions), while accounting for the phylogeny. We retained as best models all models within 2 AICc units of the minimal AICc value. Following this procedure, eight models were retained (see below).

Mean transmittance depends mainly on scale type, scale density and nanostructure density, and to a lesser extent on membrane thickness and scale colour (*Figure 5A*, *Supplementary file 1e*). The effect of scale type is retained in all eight models and is significant in all of them. Wings covered with piliform scales transmit more light than those covered with lamellar scales (*Figure 5B*). Among wings covered with lamellar scales, those with erected scales transmit more light than those with

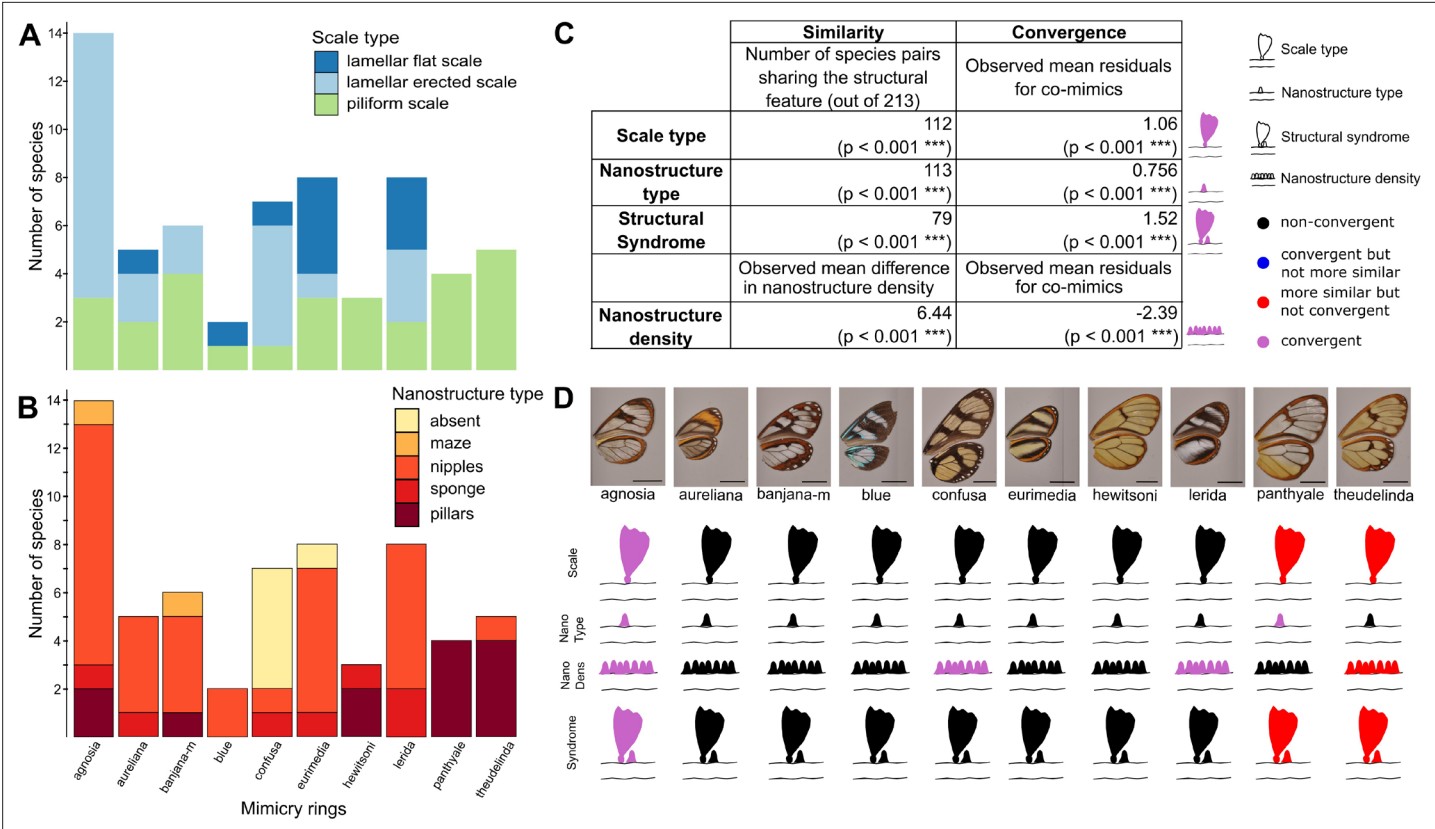

**Figure 4.** Convergence of structures underlying transparency. (**A,B**) Distribution of micro- (**A**) and nanostructures (**B**) among the different mimicry rings (indicated at the bottom on panel **B**). (**C**) Results of the test of convergence for structural features (either scale type, nanostructure type, nanostructure density or structural syndrome, that is, the association of scale type and nanostructure type). To test for similarity independently of the underlying process, we assessed whether the number of co-mimetic species sharing the same structural feature (out of 213 co-mimetic species pairs) was higher than expected at random or whether difference in nanostructure density was smaller than expected at random. To do so, we randomised 10,000 times the sharing variable (or difference in nanostructure density) over all pairs of species and we calculate the p-value (indicated in brackets, corrected for multiple testing with the 'Holm' method) as the proportion of randomisations where the number of co-mimetic species sharing the structural feature is higher than the observed number of co-mimetic species pairs sharing the structural feature (or where the mean difference in nanostructure density is smaller than the observed mean difference in nanostructure density). To test for convergence on structural features, we tested whether the observed mean residuals of the generalised linear model linking structure sharing and phylogenetic distance was higher than expected given the phylogeny (or whether mean residuals of the linear model linking difference in nanostructure density and phylogenetic distance was smaller than expected given the phylogeny) and we calculated the p-value (indicated in brackets, corrected for multiple testing with the 'Holm' method) as the proportion of randomisations of model's residuals where the mean residuals for co-mimetic species is higher (or smaller for nanostructure density) than the observed mean residuals for co-mimetic species. (**D**) Graphical representation of the results of the test of convergence for each mimicry ring. For each mimicry ring, we tested whether the structural features were more similar than expected at random and given the phylogeny (with the same tests described above, see *Figure 4—figure supplement 1* for details). We represented the results for scale type, nanostructure type, nanostructure density and structural syndrome. Black structures indicate neither more similar structures than expected at random nor convergent structures; red structures indicate structure more similar than expected at random but not convergence; blue structures indicate structures not more similar than expected at random but convergent and purple structures indicate convergent structures.

The online version of this article includes the following figure supplement(s) for figure 4:

**Figure supplement 1.** Results of the test of convergence of structural features for each mimicry ring.

flat scales. The effect of scale density is retained in the eight best models and is significant in five of those (*Supplementary file 1e*): mean transmittance decreases as scale density increases. The effect of nanostructure density is retained in six models and is significant in four of those: mean transmittance increases when nanostructure density increases (*Figure 5C*).

The interaction between scale density and nanostructure density is retained in three out of eight models and it is marginally significantly different from zero in two of those three models (*Supplementary file 1e*). The coefficient is always negative, meaning that the increase in light transmission due

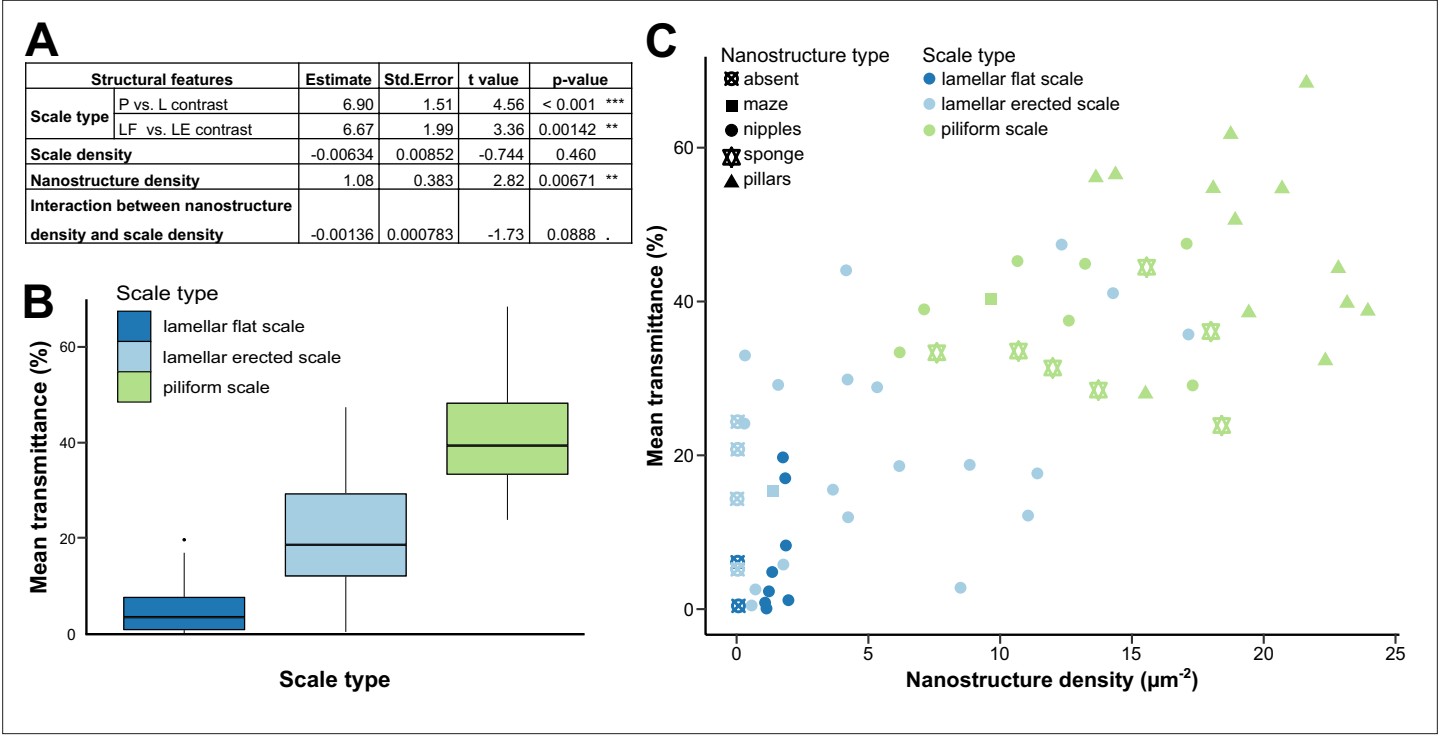

**Figure 5.** Link between mean transmittance over 300–700 nm and structural features. (**A**) Results of the best PGLS model ($F_{5,56}$ = 26.65 (p-value < 0.001 ***), AICc = 469.9, $R_{adj}^2$ = 0.678, $\lambda$ = 0 (p-value < 0.001 ***)) linking mean transmittance and micro- and nanostructure features. The explicative variables have not been scaled or centred. Nanostructure density has been measured in $\mu m^{-2}$ and scale density in $mm^{-2}$. Scale type is a categorical variable with three levels either lamellar flat scale (LF), lamellar erected scale (LE) or piliform scale (P). (**B**) Link between mean transmittance measured between 300 and 700 nm and scale type. (**C**) Link between mean transmittance measured between 300 and 700 nm and nanostructure density, nanostructure type (represented by different shapes) and scale type (represented by different colours). NB. We considered the spot corresponding to the location of the SEM images for mean transmittance.

The online version of this article includes the following source data and figure supplement(s) for figure 5:

**Source data 1.** Raw transmittance spectra presented by mimicry ring and for each species.

**Figure supplement 1.** Gradient of transparency among the studied species as illustrated by 11 species.

to the increase in nanostructure density is not as strong when scale density is high than when scale density is low. This suggests that the contribution of nanostructures to transparency is stronger when scale density is low.

The effect of membrane thickness is retained in three out of eight models and is significantly different from zero in one of them: light transmission decreases when membrane thickness increases.

Transparent scales, which do not contain pigments, transmit more light than coloured ones, which contain pigments; a relationship which is retained in three out of eight models and is marginally significantly different from zero in one model (*Supplementary file 1e*).

Other variables that were included in the model (scale length and width, nanostructure type, the interaction between scale type and scale density and the triple interaction between scale length, width and density) are not retained in any models (*Supplementary file 1e*). These results suggest that those variables do not have any strong effect on transparency.

## Discussion

We conducted the first comparative study on transparent aposematic mimetic Lepidoptera to assess whether transparency is involved in the aposematic signal, to uncover the diversity in structures underlying transparency and to assess the link between transparency and structural features.

## Convergence of transmission properties

Based on bird vision modelling applied to light transmission measurements, we showed that predators see transparent patches of species belonging to the same mimicry ring as more similar than expected at random, and given the phylogeny. Even though this result does not hold for each mimicry ring considered is this study, likely because of a lack of statistical power and/or because some mimicry rings comprise only closely related species, our results suggest that transparent patches in co-mimetic species can be under selection for convergence, mirroring what has been shown for coloured patches in opaque species (*Bybee et al., 2012*; *Llaurens et al., 2014*; *Su et al., 2015*; *Thurman and Seymoure, 2016*). This convergent resemblance, which regards mainly the degree of transparency (a general term to refer to what extent a patch appears transparent), suggests that transparent patches might be part of the aposematic signal. Nevertheless, convergence in properties of transparent patches may also result from other selective processes. Transparency is also involved in crypsis (*Arias et al., 2020b*), even in aposematic prey (*Arias et al., 2019*; *McClure et al., 2019*), and the degree of transparency needed to achieve crypsis may depend on the ambient light (*Johnsen and Widder, 1998*; *Arias et al., 2020a*). Specifically, in bright environments only highly transparent prey are cryptic, whereas in darker environments moderately transparent prey can be cryptic. In our case, as co-mimetic species share their habitat (*Chazot et al., 2014*) and microhabitat (*Willmott et al., 2017*), characterised by a specific ambient light, we cannot rule out that the similarity in the degree of transparency observed between co-mimetic species is the result of selection for crypsis rather than aposematism. Moreover, habitat-specific conditions, such as temperature or humidity, could also affect the evolution of transparent patches. Therefore, we cannot rule out that the observed convergence is driven by such abiotic factors (related to thermoregulation for example) instead of predation pressure. However, several studies on Ithomiini butterflies have shown that multiple mimicry rings usually coexist in the same localities (*Beccaloni, 1997*; *Chazot et al., 2014*; *Elias et al., 2008*; *Willmott et al., 2017*). For example, the species belonging to the mimicry rings 'banjana-m', 'panthyale', 'hewitsoni', and 'theudelinda' are all high-altitude species that are found in the same localities, and are therefore exposed to the same environmental conditions (ambient light, temperature, humidity). The fact that co-occurring chemically defended species that belong to different mimicry rings differ in transmission properties of their transparent patches (*Figure 1—figure supplement 3*) suggests that the convergence observed is likely driven by Müllerian mimicry, and is not only the result of selection for crypsis or local adaptation to abiotic factors.

This study therefore challenges our vision of transparency, which might have evolved under multiple selective pressures in aposematic butterflies. Transparency has been shown to be involved in camouflage and to decrease detectability by predators (*Arias et al., 2020b*), even in aposematic species (*Arias et al., 2019*). Yet, our results suggest that transparent patches might also participate in the aposematic signal and that selection acts on the transmission properties of these patches, particularly on the degree of transparency, but also on chromatic aspects to some extent. Therefore, transparent aposematic Lepidoptera benefit from a double protection from predation, which can act at different distances (*Barnett et al., 2018*; *Cuthill, 2019*; *Tullberg et al., 2005*): transparent aposematic species are less detectable than opaque species (*McClure et al., 2019*), but when detected they may be recognized as unpalatable by experienced predators, due to their aposematic wing pattern, and spared by those predators.

## Structural features underlying transparency

### Diversity of structures underlying transparency

We revealed an unexpected diversity of structures underlying transparency. Across the 62 species of the study, we found different microstructures in the transparent patches: transparent and coloured flat scales, transparent and coloured erected scales and piliform scales. Forked piliform scales have previously been reported in the highly transparent nymphalid species *Greta oto* (*Binetti et al., 2009*; *Siddique et al., 2015*), which belongs to the mimetic butterfly Ithomiini tribe. Erected scales (i.e. with a non-zero angle between the scale basis and the wing membrane) have been previously reported in the riodinid *Chorinea sylphina* (*Dushkina et al., 2017*) and in the nymphalid *Parantica sita* (*Goodwyn et al., 2009*). Here, we describe some species with coloured erected scales that are completely perpendicular to the wing membrane, such as in the ithomiine *Methona curvifascia*, and

some species with transparent scales. Both these features were reported in transparent Lepidoptera only recently (*Gomez et al., 2021*). Other means of achieving transparency reported in the literature are not observed among our species (e.g. wing membrane devoid of scales, *Yoshida et al., 1996*; *Gomez et al., 2021*). However, unlike *Gomez et al., 2021*, who studied a large number of transparent species with a wide range of ecologies and belonging to 31 families, our study is restricted to mimetic transparent butterflies and, as such, spans a relatively small number of families. Scales in Lepidoptera are not only involved in colour patterns but also play a role in hydrophobicity. The scale modifications underlying transparency described in our study may impair the waterproofing properties of wings, as shown by *Goodwyn et al., 2009* the wing of the translucent papilionid *Parnassius glacialis* are less hydrophobic than most Lepidoptera wings. Transparency may therefore come at a cost, especially for tropical Lepidoptera living in humid environments.

We also revealed an unexpected diversity of nanostructures covering the wing membrane, which we classify into five categories: absence of nanostructures, maze-like, nipple arrays, sponge-like and pillar-shaped nanostructures. While nipple arrays and pillars have previously been described on the wing of the sphingid *Cephonodes hylas* (*Yoshida et al., 1997*) and in the nymphalid *Greta oto* (*Binetti et al., 2009*; *Siddique et al., 2015*), respectively, maze-like nanostructures have only been reported on the corneal surface of insect eyes (*Blagodatski et al., 2015*). Moreover, the sponge-like type of nanostructures is reported here for the first time. Those nanostructures can be related to the classification proposed by *Blagodatski et al., 2015*: pillars are a subcategory of nipple arrays, with higher and more densely packed nipples with enlarged bases; sponge-like nanostructures are similar to dimples (holes embedded in a matrix), although with much bigger and more profound holes. Nipples, mazes and dimples have been found to be produced by Turing's reaction-diffusion models, a solid framework that explains pattern formation in biology (*Turing, 1952*). Theoretical models of nanostructure formation in a tri-dimensional space and developmental studies are needed to understand the process by which nanostructures are laid on butterfly wing membranes (*Pomerantz et al., 2021*).

## Link between structural features and transmission properties

The diversity of structures underlying transparency described above raises the question of whether these different structures confer different visual aspects. We indeed showed that mean transmittance over 300–700 nm, which is a proxy of the degree of transparency, depends on several structural features: scale type, scale density, nanostructure density, wing membrane thickness and scale colour. To summarise, mean transmittance increases when membrane coverage decreases, either due to reduced scale surface and/or scale density, because there is less material interacting (reflecting, diffusing, or absorbing) with light. Mean transmittance also increases when nanostructure density increases. Light transmission is indeed negatively correlated to light reflection and nanostructures are known to have anti-reflective properties, as demonstrated in the sphingid *Cephonodes hylas* (*Yoshida et al., 1997*) and in the nymphalid *Greta oto* (*Siddique et al., 2015*). Reflection increases as the difference in refractive index between air and organic materials increases. Nanostructures create a gradient of refractive index between air and wing tissue, and gradient efficiency in reducing reflection increases with a smooth increase in proportion of chitin inside the nanostructures. For instance, pillars with conical bases are more effective at cancelling reflection than cylinders because cones produce a smoother air:chitin gradient from air to wing than cylinders (*Siddique et al., 2015*). Nanostructure shape is thus important in creating a smooth gradient. In our case, nanostructure density is highly correlated to nanostructure type, which we have defined according to their shape (phylogenetic ANOVA on nanostructure density with nanostructure type as factor: $F = 26.26$, p-value = 0.001, see *Supplementary file 3d and e* for details). Specifically, the nanostructures whose shape likely creates the smoother gradient (pillar and sponge) are also the denser ones. This can explain why nanostructure type is not retained in our models because variation in mean transmittance is already explained by nanostructure density, a quantitative variable. When nanostructure density increases, light reflection thus decreases. Light can either be transmitted, reflected or absorbed, and assuming that the chitin wing membrane only absorbs a small amount of light between 300 and 700 nm (*Stavenga et al., 2014*), when light reflection decreases because of the presence of nanostructures light transmission necessarily increases, which explains the positive effect of nanostructure density on mean transmittance.

We showed that mean transmittance decreases when membrane thickness increases, because wing membrane is mainly made of chitin and even if chitin absorbs a little amount of light (*Stavenga et al.,*

*2014*), thicker membranes, which contain more chitin, absorb more light than thinner ones, thereby reducing light transmission.

We finally showed, as *Gomez et al., 2021*, that wings covered with transparent scales transmit more light than wings covered with coloured scales. This is due to the presence of pigments, such as melanins or ommochromes commonly found in butterfly scales, which absorb some part of the light spectrum, thereby reducing light transmission.

Given the high structural diversity uncovered, future studies should thoroughly quantify the relative contributions of micro and nanostructures on the produced optical effects, notably on reflection in transparent patches, which may encourage bio-inspired applications for transparent materials.

## Selection on optical properties as a potential driver of the evolution of structures

We showed that transmission properties are convergent among co-mimetic species and that they depend on the underlying structural features, which confer peculiar visual aspects, raising the question of the putative convergence of structural features among co-mimetic species. We indeed showed that despite the high phylogenetic signal of structures underlying transparency that points to the existence of developmental constraints, both micro- and nanostructural features are convergent among co-mimetic species for some mimicry rings. Convergence is also detected for structural syndrome (i.e. association between micro- and nanostructures). Our data suggest that nanostructures are more labile than microstructures. Nanostructures could therefore evolve more readily in response to selection on the degree of transparency. We showed that the presence and higher densities of nanostructures increase mean transmittance when scale density is already low, thereby allowing fine-tuning of transparency. The interplay between scales and nanostructures can thus modulate the degree of transparency and the selective pressures on the transmission properties of transparent patches may select specific associations of structural features.

To conclude, this study reveals convergence of transparency features in aposematic mimetic Lepidoptera, which may be the result of selection by predators, likely through aposematism, even though transparent patches may also be under other local selection pressures such as selection for crypsis or adaptation to climatic conditions. Transparency entails strong structural modifications of scales that might impair other functions such as thermoregulation (*Berthier, 2005*), hydrophobicity (*Goodwyn et al., 2009*) and perhaps mate signalling. Transparency may therefore come at a cost in those large-winged insects, which may explain why it is not pervasive among Lepidoptera.

# Materials and methods

For further details about materials and methods see the Materials and methods section in the Appendix.

## Material

In this study, we focus on 62 different species represented by one or two specimens collected with hand nets in understory forests in Peru and Ecuador, by ourselves and private collectors (*Supplementary file 1d*). The choice of species (and therefore the sample size) was dictated by the availability of specimens that could be imaged in SEM, and therefore destroyed (which precludes using collection specimens). We attempted to maximise the number of mimicry rings, the number of species within mimicry rings, and the phylogenetic diversity within mimicry rings. The selected species belong to seven different families (Nymphalidae, Riodinidae, Pieridae, Papilionidae, Erebidae, Notodontidae, Geometridae) and represent 10 different mimicry rings, following the classification used in Ithomiini: 'agnosia', 'aureliana', 'banjana-m', 'confusa', 'eurimedia', 'hewitsoni', 'lerida', 'panthyale', 'theude-linda' (*Chazot et al., 2014*; *Willmott et al., 2017*; *Willmott and Mallet, 2004*). In addition, we call 'blue' a mimicry ring that does not include Ithomiini species. While most of these species are transparent to some extent, some of them are opaque or nearly so, but still resemble clearwing species (see *Figure 5—figure supplement 1*).

## Phylogeny

We used both published and de novo (see 'Phylogeny' section in SI for detailed protocol) sequences from one mitochondrial gene and seven nuclear genes, representing a total length of 7433 bp to infer a molecular phylogeny (knowing that for many taxa there are missing data, see *Supplementary file 2a*). To improve the phylogeny topology, we added 35 species representing eight additional families to the dataset (see *Supplementary file 2a*). We performed a Bayesian inference of the phylogeny using BEAST 1.8.3 (*Baele et al., 2017*). We forced the monophyly of some groups and we added eleven secondary calibration points (see *Supplementary file 2b*) following *Kawahara et al., 2019*.

## Spectrophotometry

Specular transmittance was measured over 300–700 nm, a range of wavelengths to which both birds and butterflies are sensitive (*Briscoe and Chittka, 2001*; *Hart, 2001*) using a custom-built spectrophotometer (see 'Spectrophotometry' section in SI for details). For each species, we measured five different spots in the transparent patches on the ventral side of the forewing (see *Figure 1* for location). We computed mean transmittance over 300–700 nm from smoothed spectra using pavo (*Maia et al., 2019*), as a proxy for transparency: wing transparency increases as mean transmittance increases. On a subset of 16 species, we measured 2–3 specimens per species and given that measurements were repeatable (see 'Spectrophotometry' section in SI), we retained only one specimen per species for optical measurements.

## High-resolution imaging and structure characterisation

We observed structures with a digital photonic microscope (Keyence VHX-5000) to determine scale form (lamellar scale vs. piliform scale), scale colour (coloured vs. transparent) and scale insertion (flat vs. erected) on ventral side, which is the side exposed at rest for most of the species in this study. Moreover, we checked that there were not significant differences between ventral and dorsal sides regarding main structural features (see Appendix and *Supplementary file 3f,g and h*). We defined as scale type the interaction between scale form and scale insertion (erected lamellar scale, flat lamellar scale and piliform scale). Wings were imaged using SEM (Zeiss Auriga 40) to determine nanostructure type and to measure scale density, scale length and width, membrane thickness, and nanostructure density (see SI for more details). We also determined for each species the structural syndrome, defined as the association between micro- and nanostructural features. On a subset of 3 species, we measured 10 specimens per species, each specimen being measured twice for density and five times for scale dimensions. Given that scale structural features were shown to be repeatable (see 'High-resolution imaging and structure characterisation' section in SI) within species we retained one specimen per species in structure characterisation.

## Vision models

We used bird vision modelling on the smoothed transmission spectra to test whether transparent patches of co-mimetic species are perceived as similar by birds. Birds differ in their sensitivity to UV wavelength: some are more sensitive to UV (UVS vision) than others (VS vision). As predators of neotropical butterflies can belong to either category (*Dell'Aglio et al., 2018*), we used wedge-tailed shearwater (*Puffinus pacificus*) as a model for VS vision (*Hart, 2004*) and blue tit (*Cyanistes caeruleus*) as model for UVS vision (*Hart et al., 2000*). We considered two different light environments differing in their intensity and spectral distribution: forest shade and large gap as defined by *Endler, 1993*; *Gomez and Théry, 2007*. In our model, we considered that the butterfly was seen against the sky (light is just transmitted through the wing). We used the receptor-noise limited model of *Vorobyev and Osorio, 1998* with neural noise and with the following relative cone densities 1:1.9:2.7:2.7 (for UVS:S:M:L, *Hart et al., 2000*) and 1:0.7:1:1.4 (for VS:S:M:L, *Hart, 2004*) for UVS and VS vision respectively, and a Weber fraction of 0.1 for chromatic vision (*Lind et al., 2013a*; *Maier and Bowmaker, 1993*) and 0.2 for achromatic vision (average of the two species studied in *Lind et al., 2013b*) for both visual systems to compute chromatic and achromatic contrasts. In total we calculated four different vision models, using the R package pavo (*Maia et al., 2019*), representing all combinations of bird visual systems and light environments.

We extracted the chromatic and achromatic contrasts between each pair of species in the dataset, comparing only analogous spots (i.e. occupying the same position) on the forewing.

## Statistical analyses

All statistical analyses were performed with the software R version 3.6.2 and 4.0.3 (*R Development Core Team, 2019*). All scripts and data used to produce the results of statistical analyses are available at *Pinna, 2021*, https://github.com/ChPinna/Lepidoptera_Transparency-mimicry; copy archived at swh:1:rev:fb5017880f034cfd818d7f5f5f4acc51530680fb.

## Convergence on optical properties

To assess whether transparent patches, as perceived by predators, were more similar than expected at random, we calculated the mean phenotypic distance (either chromatic or achromatic contrast) for co-mimetic species and we compared this mean phenotypic distance to a null distribution of this mean distance, where the phenotypic distance has been randomised 10,000 times over the 1891 possible pairs of species, irrespective of their phylogenetic relationship. The p-value was calculated as the proportion of randomisations where the calculated mean distance for co-mimetic species was smaller than the observed mean distance. The result of this test allows us to determine whether co-mimetic species are perceived as similar by their main predators, irrespectively of the evolutionary underlying mechanism, which can be either shared ancestry of convergent evolution. To disentangle the two possible mechanisms, we accounted for the phylogenetic relationship between species by performing a linear regression between phenotypic distances and phylogenetic distances for each pair of species, following *Elias et al., 2008*. Pairs of species below the regression line (with a negative residual) are phenotypically more similar than expected given the phylogeny. To test whether pairs of co-mimetic species were mostly below the regression line, we calculated the observed mean residuals for co-mimetic species and we compared it to a null distribution of mean residuals for co-mimetic species, where residuals have been randomised 10,000 times over the 1891 possible pairs of species. The p-value was calculated as the proportion of randomisations where the calculated mean residuals were smaller than the observed mean residuals for co-mimetic species. We also tested for each mimicry ring whether co-mimetic species were perceived as more similar as expected at random and given the phylogeny by applying the tests described above as follows: we calculated mean phenotypic distance and mean residuals, respectively, for pairs of species belonging to the considered mimicry ring and compared these means to the random distribution of phenotypic distance and residuals, respectively, of the model restricted to the same number of observations (i.e. pair species) as in the mimicry ring considered. For each series of tests (i.e. with and without phylogenetic correction and for each spot in each vision model) we applied a correction for multiple testing using the 'Holm' method.

## Phylogenetic signal

To assess whether transmission properties and structural features were conserved in the phylogeny, we estimated the phylogenetic signal of each variable. For quantitative variable (mean transmittance, scale density, scale length, scale width, nanostructure density, and membrane thickness), we calculated both Pagel's $\lambda$ (*Pagel, 1999*) and Blomberg's K (*Blomberg et al., 2003*) implemented in the R package 'phytools' (*Revell, 2012*). For multicategorical variables (scale type and nanostructure type), we used the δ-statistic (*Borges et al., 2019*) and we compared it to the distribution of values of δ when the trait is randomised along the phylogeny to estimate whether the trait is randomly distributed along the phylogeny. Finally, for binary variables (scale colour), we used Fritz and Purvis' D (*Fritz and Purvis, 2010*) implemented in the R package 'caper' (*Orme et al., 2018*).

## Convergence on structures

We tested whether structural features (microstructures, i.e. scales, nanostructures, and nanostructure density but also structural syndrome, that is the association between microstructures and nanostructures) are more similar between co-mimetic species than expected at random. To do so, we considered every pair of species in our dataset and we calculated the number of co-mimetic species sharing the same structural features. We compared this number to the null distribution of the number of species sharing the same structural features where the structural feature has been randomised 10,000 times, a method similar to that used in *Willmott and Mallet, 2004*. We calculated the p-value as the proportion of randomisations where the number of species sharing structures is higher than the observed number. For nanostructure density, we calculated the difference in nanostructure density for each pair of species and we calculated the mean difference in nanostructure density for co-mimetic species.

We then randomised these differences 10,000 times and we calculated a p-value as the proportion of randomisations where the mean nanostructure density between co-mimetic species is smaller than the observed mean nanostructure density. To determine whether this sharing of structures was due to convergent evolution, we performed a generalised linear model with a binomial error distribution linking the variable for structure sharing (one if species shared the same structure, 0 otherwise) with phylogenetic distance. We then calculated the mean residuals of the model for co-mimetic species and we compared it to a null distribution of mean residuals for co-mimetic species, where residuals have been randomised 10,000 times. The p-value was given by the proportion of randomisations where the calculated mean residuals for co-mimetic species was higher than the observed mean residuals for co-mimetic species. For nanostructure density, we performed a linear model linking differences in nanostructure density with phylogenetic distances. We calculated the mean residuals of the model for co-mimetic species and we randomised residuals 10,000 times. We calculated p-value as the proportion of randomisations where mean residuals for co-mimetic species is smaller than the observed mean residuals. We also tested for each mimicry ring whether co-mimetic species share structural features more than expected at random and given the phylogeny by applying the tests described above as follows: we calculated either the number of species sharing the same structural feature or the mean phenotypic distance and mean residuals, respectively, for pairs of species belonging to the considered mimicry ring and compared these means to the random distribution of the number of species sharing the same structural feature of the mean phenotypic distance and residuals, respectively, of the model restricted to the same number of observations (i.e. pair species) as in the mimicry ring considered. For each series of tests (i.e. with and without phylogenetic correction and for each structures), we applied a correction for multiple testing using the 'Holm' method.

## Link between transparency (mean transmittance) and structures

To assess the link between structural features and the degree of transparency we only used the spectrophotometric data of the points that correspond to the location of the SEM images (between 1 and 3 points per species) and we calculated the average of mean transmittance over 300–700 nm for each specimen (see *Supplementary file 1d*). We tested the link between this average mean transmittance and all the structural features we measured (scale type, scale colour, scale density, scale length, scale width, nanostructure type, nanostructure density, membrane thickness and the following interactions: interaction between scale type and scale density, interaction between scale density and nanostructure density and the triple interaction between scale density, scale length and scale width), while controlling for phylogenetic relationships by performing Phylogenetic Generalised Least Square regression (PGLS) implemented in the R package 'caper' (*Orme et al., 2018*). We compared all possible models with all the structural variables, but we prevented some variables from being in the same model because they were highly correlated, using the R package 'MuMIn' (*Barton, 2019*). Among the 308 models, we selected the best models (difference in AICc inferior to 2). Eight such models were retained.

## Acknowledgements

We thank Jonathan Pairraire and Céline Houssin for the help with photonic imaging and Josquin Gerber and Edgar Attivissimo for helping with spectroscopic measurements. We thank Benoit Vincent for the identification of some Arctiini specimens. We thank Keith Willmott, Raúl Aldaz, Paola Santacruz Endara, José Simbaña, Ramón Mamallacta, Alexandre Toporov, Stéphanie Galusser, Jay, Bruno and César Ramirez Garcia for companionship and help in the field over the last 15 years, and Gerardo Lamas, Luis Figueroa and Santiago Villamarín for assistance with research permits. We are thankful to the Institut de Physique du Globe de Paris (IPGP) for giving us access to SEM and to the Peruvian and Ecuadorian governmental authorities for collection permits (021 C/C-2005-INRENA-IANP, 002–2015-SERFOR-DGGSPFFS, 373–2017-SERFOR-DGGSPFFS, 005-IC-FAU-DNBAPVS/MA, 019-IC-FAU-DNBAPVS/MA). This work was funded by Clearwing ANR project (ANR-16-CE02-0012), HFSP project on transparency (RGP0014/2016) and a France-Berkeley fund grant (FBF #2015-58).

# Additional information

## Funding

| Funder | Grant reference number | Author |
|---|---|---|
| Human Frontier Science Program | RGP0014/2016 | Nipam H Patel<br>Serge Berthier<br>Marianne Elias |
| Agence Nationale de la Recherche | ANR-16-CE02-0012 | Serge Berthier<br>Christine Andraud<br>Doris Gomez<br>Marianne Elias |
| France-Berkeley fund grant | FBF #2015-58 | Nipam H Patel<br>Marianne Elias |

The funders had no role in study design, data collection and interpretation, or the decision to submit the work for publication.

## Author contributions

Charline Sophie Pinna, Data curation, Formal analysis, Investigation, Methodology, Visualization, Writing – original draft, Writing – review and editing; Maëlle Vilbert, Investigation, Methodology, Validation, Writing – review and editing; Stephan Borensztajn, Data curation, Methodology, Validation; Willy Daney de Marcillac, Methodology, Validation; Florence Piron-Prunier, Investigation, Writing – review and editing; Aaron Pomerantz, Continuous discussion throughout the data acquisition, analyses and writing, Resources, Writing – review and editing; Nipam H Patel, Funding acquisition, Resources, Writing – review and editing; Serge Berthier, Funding acquisition, Methodology, Supervision, Writing – review and editing; Christine Andraud, Investigation, Methodology, Supervision, Writing – review and editing; Doris Gomez, Conceptualization, Funding acquisition, Investigation, Methodology, Project administration, Resources, Supervision, Writing – review and editing; Marianne Elias, Conceptualization, Data curation, Funding acquisition, Investigation, Methodology, Project administration, Resources, Supervision, Writing – review and editing

## Author ORCIDs

Charline Sophie Pinna (ID) http://orcid.org/0000-0002-4947-0893
Florence Piron-Prunier (ID) http://orcid.org/0000-0003-2652-5123
Aaron Pomerantz (ID) http://orcid.org/0000-0002-6412-9001
Nipam H Patel (ID) http://orcid.org/0000-0003-4328-654X
Serge Berthier (ID) http://orcid.org/0000-0002-9255-059X
Christine Andraud (ID) http://orcid.org/0000-0002-3112-9363
Doris Gomez (ID) http://orcid.org/0000-0002-9144-3426
Marianne Elias (ID) http://orcid.org/0000-0002-1250-2353

## Decision letter and Author response

Decision letter https://doi.org/10.7554/eLife.69080.sa1
Author response https://doi.org/10.7554/eLife.69080.sa2

# Additional files

## Supplementary files

• Supplementary file 1. Supplementary files related to transparent wing optical properties and structures. (a) Tests of convergence of transparent patches, as perceived by predators, among co-mimetic species: achromatic contrasts. All the visual systems (VS and UVS) and the illuminants (large gap 'lg' and forest shade 'fs') tested are presented. We tested whether mean achromatic contrast (dL) between co-mimetic species is smaller than expected at random (expected mean dL for co-mimics ± standard deviation (sd)). To do so, we randomised the value of the achromatic contrast 10,000 times over each pair of species and we calculated the p-value as the proportion of randomisations where the mean achromatic contrast for co-mimics is smaller than the observed mean achromatic contrast. We also considered whether co-mimetic species were more similar than expected according to their phylogenetic relationship. To do so, we did a linear model between

achromatic contrasts and phylogenetic distances to account for the effect of phylogeny on the achromatic contrast and we considered the mean of residuals for co-mimetic species. If co-mimetic species are more similar than expected according to their phylogenetic relationship, the mean of residuals should be negative. To test whether the mean of residuals is smaller than expected according to the phylogeny, we randomised residuals over all pair of species and we calculated the mean of residuals for co-mimetic species. We calculated 'p-value with phylogenetic correction' as the proportion of randomisations where the mean of residuals is smaller than the observed mean of residuals. If the p-value is smaller than 0.05, it means that co-mimetic species are more similar than expected at random and if the p-value with phylogenetic correction is smaller than 0.05, it means that the observed similarity is due to convergence. We also present p-values corrected with multiple testing with the 'Holm' method. (b) Tests of convergence of transparent patches, as perceived by predators, among co-mimetic species: chromatic contrasts. All the visual systems (VS and UVS) and the illuminants (large gap 'lg' and forest shade 'fs') tested are presented. We tested whether mean chromatic contrast (dS) between co-mimetic species is smaller than expected at random (expected mean dS for co-mimics ± standard deviation (sd)). To do so, we randomised the value of the chromatic contrast 10,000 times over each pair of species and we calculated the p-value as the proportion of randomisations where the mean chromatic contrast for co-mimics is smaller than the observed mean chromatic contrast. We also considered whether co-mimetic species were more similar than expected according to their phylogenetic relationship. To do so, we did a linear model between chromatic contrasts and phylogenetic distances to account for the effect of phylogeny on the chromatic contrast and we considered the mean of residuals for co-mimetic species. If co-mimetic species are more similar than expected according to their phylogenetic relationship, the mean of residuals should be negative. To test whether the mean of residuals is smaller than expected according to the phylogeny, we randomised residuals over all pair of species and we calculated the mean of residuals for co-mimetic species. We calculated 'p-value with phylogenetic correction' as the proportion of randomisations where the mean of residuals is smaller than the observed mean of residuals. If the p-value is smaller than 0.05, it means that co-mimetic species are more similar than expected at random and if the p-value with phylogenetic correction is smaller than 0.05, it means that the observed similarity is due to convergence. We also presented p-values corrected with multiple testing with the 'Holm' method. (c) Phylogenetic signal for structural features and transmission properties. Measure of the phylogenetic signal (estimated as Pagel's $\lambda$ and Blomberg's K for quantitative traits; δ for multicategorial traits and Purvis and Fritz's D for binary traits) of the different features associated to micro- and nanostructures and of mean transmittance. When $\lambda$ or K are equal to 0, the trait is distributed randomly across the phylogeny, whereas when $\lambda$ or K are equal to one the trait evolves according to a Brownian motion model along the phylogeny. When D is equal to 1, the trait is randomly distributed across the phylogeny whereas when D is equal to 0, the trait evolves according to Brownian motion model along the phylogeny. The value of δ can be any positive real number and the higher this value, the higher the phylogenetic signal of the trait. For δ, to determine whether the distribution of the trait is different from a random distribution we randomised the trait 1000 times along the phylogeny, and we calculated δ for each randomisation. We then compared the value of δ to the distribution of values of δ under the random hypothesis and we calculated a p-value as the number of randomisations in which δ is higher than the value obtained for the real distribution of the trait. (d) Information about specimens used for optical and structural measurements. (e) Results of the eight best PGLS (Phylogenetic Generalised Least Square) models (AICc within an interval of 2 of that of the best model). For each model, we give: the F statistic with the degrees of freedom in indices, the p-value of the model (in brackets), the corrected Akaike criterion (AICc) of the model, the adjusted $R^2$ and the value of lambda branch length transformation which has been estimated by maximum likelihood given the statistical model linking traits. When $\lambda$ equals 1, the branch length of the phylogeny is unchanged, whereas when $\lambda$ equals 0 branch length is set to zero, meaning that all species are considered independent. The 'p-values' for the value of $\lambda$, given in brackets, are the probability that $\lambda$ is equal to 0 or to 1. We also give for each model the value of the coefficient estimate for each variable tested and the p-value (in brackets) is represented with the follow symbols: '***': $p < 0.001$; '**': $p < 0.01$; '*': $p < 0.05$; '.' : $p < 0.1$; 'n.s.': not significantly different from 0. NA means that the variable was not retained in the model. (f) Technical repeatability of transmission measurements and structural features. For each grouping factor (either the number of species or the number of individuals or the total number of different spots measured; indicated in the 'number of groups' column), we calculated the value of repeatability R based on several measurements of the same element of a grouping factor. The calculation of repeatability is based on mixed linear models. Confidence intervals are calculated with parametric bootstraping and p-values (associated to the test $R > 0$) are

calculated with two methods: with likelihood ratio test comparing the likelihood of the model with and without the tested random effect and with permutation tests. We also calculated the coefficient of variation (CV, as the mean of the group devided by the standard error) for each group and we give here the median value of the CV distribution. (g) Biological repeatability of transmission measurements and structural features. For each grouping factor (either the number of species, or the number of different spots measured per species; indicated in the 'number of groups' column), we calculated the value of repeatability R based on several measurements of the same element of a grouping factor. The calculation of repeatability is based on mixed linear model. Confidence intervals are calculated with parametric bootstraping and p-values (associated to the test $R > 0$) are calculated with two methods: with likelihood ratio test comparing the likelihood of the model with and without the tested random effect and with permutation tests. We also calculated the coefficient of variation (CV, as the mean of the group devided by the standard error) for each group and we give here the median value of the CV distribution. (h) Similarity between conspecific individuals for chromatic and achromatic contrasts. To test whether conspecific individuals were perceived as more similar than expected at random for each spot on the forewing, we randomised the contrasts over all pair of species and we calculated the mean distance for conspecific individuals. We compared the mean phenotypic distance (either chromatic or achromatic contrast) for the observed data to the distribution of mean phenotypic distance calculated for 10,000 randomisations and we calculated the p-value as the number of randomisations where mean phenotypic distance was smaller than the observed phenotypic distance. We conclude that conspecific individuals are perceived as more similar than expected at random, implying that any individual is representative of its species.

• Supplementary file 2. Supplementary files related to phylogenetic reconstruction. (a) Information on specimens used to infer a phylogeny. (b) Node constraints used to calibrate the phylogeny. For all constraints, we used uniform distribution priors whose bound were determined according to 95 % HSPD inferred by *Kawahara et al., 2019* on their phylogeny of Lepidoptera. (c) Results of the best partition (based on BIC) for the eight different genes obtained with Partition Finder v1.0.1, with linked branch length and greedy algorithm. For each gene, pos1, pos2 and pos3 refer to codon positions. Only the substitution models available in BEAST were tested. GTR: general time reversible (base frequencies are variable, substitution matrix is symmetrical), HKY: Hasegawa-Kishino-Yano (base frequencies are variable, there are one transition rate and one tranversion rate), TrN: Tamura-Nei (base frequencies are variable, transversion rates are equal, transition rates are variable), I: proportion of invariable sites, G: gamma distribution (rate variation among sites is gamma distributed).

• Supplementary file 3. Supplementary files related to supplementary results: link between bird perception and optical properties; link between nanostructure type and density;comparision between wing ventral and dorsal sides. (a) Results of the linear mixed model linking mean transmittance over 300–700 nm (physical descriptor of transparency) and coordinates in tetrahedral colour space (x, y, z) and luminance extracted from the vision model with UVS visual system and large gap ambient light (biologically relevant descriptors of transparency). For each species, the five measurements were used, and the specimen was taken as random effect, and all the variables were centred and scaled. (b) Analysis of deviance table to check the importance of the effect of each variable on mean transmittance over 300–700 nm. (c) Results of the PGLS model ($F_{4,57} = 3728$ (p-value < 0.001 ***), AICc = –159.4, $R_{adj}^2 = 0.9959$, $\lambda = 0$ (probability$_{(\lambda = 1)}$ < 0.001)) linking average mean transmittance over 300–700 nm (physical descriptor of transparency) with average coordinates in tetrahedral space (x, y and z) and luminance extracted from vision models (biologically relevant descriptors of transparency). (d) Results of the phylogenetic ANOVA on nanostructure density with nanostructure type as factor. The p-value is based on simulations. (e) Results of the post-hoc tests (t values) to determine which type of nanostructures is different in density to others. p-values were corrected with Bonferroni correction and are indicated in brackets after t-values. Significant differences between nanostructure types are highlighted in bold. (f) Results of the type III analysis of variance on the model linking scale density with species and side (ventral or dorsal) to determine the effect of each variable on scale density. (g) Results of the type III analysis of variance on the model linking scale length with species, side (ventral or dorsal) and scale type (lamellar, piliform bifid or piliform monofid) to determine the effect of each variable on scale length. (h) Results of the type III analysis of variance on the model linking scale width with species, side (ventral or dorsal) and scale type (lamellar, piliform bifid or piliform monofid) to determine the effect of each variable on scale width.

• Transparent reporting form

## Data availability

All the data needed for computing the analyses are provided in the supplementary material, Dryad (for phylogenetic tree and gene aligment, accessible here: 10.5061/dryad.c2fqz617s) and GitHub repository (https://github.com/ChPinna/Lepidoptera_Transparency-mimicry copy archived at https://archive.softwareheritage.org/swh:1:rev:fb5017880f034cfd818d7f5f5f4acc51530680fb). The sequences are submitted to GenBank and the accession numbers are provided in the Supplementary file 2a. Those sequences can also be seen in the alignment deposited in Dryad.

The following dataset was generated:

| Author(s) | Year | Dataset title | Dataset URL | Database and Identifier |
|---|---|---|---|---|
| Pinna CS, Piron-Prunier F, Elias M | 2021 | Data from: Alignement and phylogenetic tree of 106 Lepidoptera | http://dx.doi.org/10.5061/dryad.c2fqz617s | Dryad Digital Repository, 10.5061/dryad.c2fqz617s |

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

## Appendix 1

## Materials and methods

### Phylogeny

We used published sequences from eight gene regions to infer a molecular phylogeny: the mitochondrial cytochrome oxidase c subunit 1 (COI) gene and the nuclear genes carbamyl-phosphate synthase II (CAD), malate dehydrogenase (MDH), elongation factor one alpha (EF-1), tektin (TKT), ribosomal protein S5 (RpS5), isocitrate dehydrogenase (IDH) and Glyceraldehyde 3-phosphate dehydrogenase (GAPDH), which represent a total length of 7433 bp (*Supplementary file 2*). To improve phylogeny topology, we added 35 species representing eight additional families to the dataset. When no sequence was available for a particular species on Genbank, we sequenced de novo the COI, CAD and MDH genes of that species (*Supplementary file 2*). We have missing data for some species, but we had at least the COI sequence for each species considered.

For de novo sequencing, DNA was extracted from butterfly legs with a DNeasy Blood & Tissue Kit (QIAGEN laboratory) and targeted genes were amplified with PCR conditions adapted from *Wahlberg and Wheat, 2008*. COI, CAD and MDH were amplified in two pieces with the primers described in *Wahlberg and Wheat, 2008*. PCR were performed in a volume of 25 μL with 2–4 μL of genomic DNA, 1 μL of each primer at a concentration of 100 pmol/μL, 1 μL of nucleotides at a concentration of 2 mM, 2.5 μL of DreamTaq buffer, 0.125 μL of DreamTaq polymerase. The elongation phase was reduced to 70 seconds. For CAD and MDH, the annealing temperature was reduced to 50 °C for most specimens. Eurofins Genomics sequenced the PCR products with Sanger method.

Sequences were aligned with CodonCodeAligner (version 3.7.1.1, CodonCode Corporation, http://www.codoncode.com/) and concatenated with PhyUtility (version 2.2, *Smith and Dunn, 2008*). The dataset was then partitioned by gene and codon positions and the best models of substitution were selected over all models implemented in BEAST, using the 'greedy' algorithm and linked rates implemented in Partition Finder 1.0.1 (*Lanfear et al., 2012*, see *Supplementary file 2c* for best scheme). We performed a Bayesian inference of the phylogeny using BEAST 1.8.3 (*Baele et al., 2017*) on the Cipres server (*Miller et al., 2010*). We constrained some clades to be monophyletic (notably Ithomiini, Danainae, Nymphalidae, Riodinidae, Pieridae, Papilionidae, Erebidae, Notodontidae, Geometridae, Noctuoidae, Papilionoidae) and we calibrated the crown age and divergence time of some groups (see *Supplementary file 2b*), following *Kawahara et al., 2019*. Four independent analyses were run for 50 million generations, with one Monte Carlo Markov chain each and a sampling frequency of one out of 50,000 generations (resulting in 1,000 posterior trees). After checking for convergence of the two best analyses, the posterior distributions of these two runs were combined (using logCombiner 1.8.2, *Drummond and Rambaut, 2007*), with a burnin of 10 %. The maximum clade credibility (MCC) tree with median node ages was computed using TreeAnnotator 1.8.2 (*Figure 3—figure supplement 1*). Species not represented in our dataset were then pruned from the tree. The MCC tree was used for subsequent phylogenetic analyses.

### Spectrophotometry

Specular transmission was measured over 300–700 nm, a range of wavelength to which both butterflies and birds, which are expected to be their main predators, are sensitive (*Briscoe and Chittka, 2001*; *Hart, 2001*) with a custom-built set-up composed of a 300 W Xenon lamp emitting light over 200–1160 nm, a collimated emitting optic fibre (UV to NIR multimod fibre with a core diameter of 50 μm) illuminating the wing sample with a 1 mm diameter spot and a collimated collecting optic fibre (Avantes UV to IR multimod fibre with a core diameter of 200 μm, FC-UVIR 200–1) connected to the spectrometer (SensLine AvaSpec-ULS2048XL-EVO, Avantes). Fibres are aligned and 22 cm apart. The wing is placed perpendicular to the fibres at equal distance with the ventral side facing the illuminating fibre. The spectrometer has a resolution of 0.5 nm and transmittance is calculated relative to a dark (light patch blocked at the end of the illuminating fibre) and to a white reference (no sample between the fibres):

$$Transmittance(\lambda) = \frac{S_{sample}(\lambda) - S_{dark}(\lambda)}{S_{reference}(\lambda) - S_{dark}(\lambda)},$$

where $\lambda$ represents the wavelength, S the number of photons counted by the spectrometer for this wavelength for sample, dark and reference measurements. Each spectrum was smoothed with the loess function using R software (version 3.6.2.) (0.2 span on 500–700 nm, and 0.05 on 300–700 nm). We computed mean transmittance over 300–700 nm (B2) using pavo (*Maia et al., 2019*), as a proxy for transparency: wing transparency increases as mean transmittance increases.

We assessed measurement repeatability (i. e., technical replication) on 11 species, representing part of six mimicry rings and belonging to six different families. We measured three times each of the five spots on the forewing (see *Figure 1* for location) for 2–3 individuals per species. We assessed measurement repeatability of mean transmittance (B2, *Montgomerie, 2006*) and of chroma (S8, *Montgomerie, 2006*) over 400–700 nm by calculating repeatability with species (11 groups for the 11 species considered), individual (32 groups for the 2–3 specimens per species) and spot (160 groups for the five spots measured for each of the 32 specimens) as random effect with rpt function from rptR package (*Stoffel et al., 2017*): $R_{species, B2} = 0.586$, p-value $< 0.001$; $R_{individual, B2} = 0.0535$, p-value $= 0.033$; $R_{spot, B2} = 0.34$, p-value $< 0.001$; $R_{species, S8} = 0.583$, p-value $< 0.001$; $R_{individual, S8} = 0.0476$, p-value $= 0.053$; $R_{spot, S8} = 0.355$, p-value $< 0.001$ (*Supplementary file 1f*).

We assessed intraspecific variation (i. e., biological replication) on 19 species, for which we had more than one individual, that represented six mimicry rings and belonged to six different families. We measured each of the five spots on the forewing once (see *Figure 1* for location) for 2–3 individuals per species. We assessed intraspecific variation of mean transmittance (B2, *Montgomerie, 2006*) and of chroma (S8, *Montgomerie, 2006*) over 400–700 nm by calculating repeatability with species (19 different groups) and spot (95 different groups corresponding to the five spot for each species) as random effect with rpt function from rptR package (*Stoffel et al., 2017*): $R_{species, B2} = 0.671$, p-value $< 0.001$; $R_{spot, B2} = 0.145$, p-value $< 0.001$; $R_{species, s8} = 0.671$, p-value $< 0.001$; $R_{spot, S8} = 0.0441$, p-value $= 0.022$ (*Supplementary file 1g*). As all measurements were repeatable, we considered that any individual is representative of its species. One specimen per species was therefore used in all analyses.

Due to technical issues, measurements for repeatability could only be performed for wavelengths ranging from 400 to 700 nm. However, both mean transmittance and chroma over 400–700 nm were highly correlated to mean transmittance and chroma over 300–700 nm, respectively (correlation coefficient equals 0.9979 [0.9974; 0.9983] (p-value $< 0.001$) for mean transmittance and 0.9765 [0.9707; 0.9812] (p-value $< 0.001$) for chroma). Repeatability in measurements over 400–700 nm can therefore be extrapolated to the full 300–700 nm range.

## High-resolution imaging and structure characterisation

Dry wings were cut from specimens before being gold-coated (10 nm thick layer) and observed in SEM (Zeiss Auriga 40). Top-view and cross section SEM images were analysed with ImageJ 1.52 (*Schindelin et al., 2012*) to measure scale density, scale length and width, membrane thickness, and nanostructure density. To measure scale density, we counted the number of scales in three different rectangular areas on a SEM photo and we calculated the mean scale density for each specimen. We measured scale length and width on three different scales, and we calculated the mean length and width for each specimen. We measured membrane thickness on five different photos per specimen and we calculated the mean membrane thickness. For nanostructure density, we coded macros (one for each type of nanostructure) in ImageJ (*Schindelin et al., 2012*) to count the number of structures on the image. For each specimen, we had two types of images with different magnifications. As the densities were congruent between the two types of images (measure of repeatability with rptR package: $R = 0.368$, p-value $< 0.001$), we calculated the mean nanostructure density between them.

We assessed intraspecific variation (i. e., biological replication) for scale characteristics (density, length and width) on three species, belonging to the mimicry ring agnosia: *Ithomia agnosia*, *Pseudoscada timna* and *Heterosais nephele*. We used 5 males and 5 females per species. We measured scale density for proximal and distal zone on forewing twice and we measured scale length and width for 10 different scales per specimen. We assessed repeatability with rptR with species as random effect (see *Supplementary file 1g and f*): for density, $R_{species} = 0.545$ (p-value

< 0.001); for scale length, $R_{species}$ = 0.241 (p-value < 0.001); for scale width, $R_{species}$ = 0.579 (p-value < 0.001). As scale structural features were repeatable within species, we used one specimen per species.

## Results

### Supplementary result 1

Link between physical and biologically relevant descriptors of transparency

We confirmed that mean transmittance is correlated to the x, y and z coordinates in tetrahedral colour space and luminance by performing a mixed linear model linking mean transmittance to the x, y and z coordinates and the luminance for each spectrum with specimen as random factor (*Supplementary file 3a*), using the lme function from the nlme R package (*Pinheiro et al., 2021*). The deviance analysis (*Supplementary file 3b*) on this linear model showed that each variable had a significant effect on mean transmittance. We also confirmed with a phylogenetic least square analysis (PGLS), which accounts for species relatedness, that average mean transmittance is correlated to average coordinates in tetrahedral space and luminance (*Supplementary file 3c*). Taken together, these results mean that what is perceived by predators is correlated to the physical property, in this case mean transmittance.

### Supplementary result 2

Structural features differences between ventral and dorsal sides

To check for differences in microstructure (i.e. scale) density and dimensions between ventral and dorsal wing sides, we carried out a preliminary study on 12 Ithomiini species (*Aeria eurimedia, Episcada hymen, Godyris dircenna, Heterosais nephele, Mcclungia cymo, Methona grandior, Pagyris cymothoe, Paititia neglecta, Pseudoscada florula, Pteronymia forsteri, Scada reckia, Thyridia psidii*, some of them absent from this study). To do so, we analysed photonic digital microscopic images from ventral and dorsal sides for each species. For scale density, we used one photo of each side and calculated density as described in the subsection 'High-resolution imaging and structure characterisation' in the 'Supplementary Materials and Methods' section (see above). For scale dimension, we used the photos of 3 scales of each type present in each species and calculated mean length and mean width.

We performed linear models to explain the variation in scale density (*Supplementary file 3f*), scale length (*Supplementary file 3g*), or scale width (*Supplementary file 3h*). We included species, wing side, scale type and their interaction as explanatory variables. For scale density we considered lamellar vs piliform scale type, whereas for scale dimension, we considered lamellar, piliform monofid and piliform bifid as scale type.

