## [Editor Report]

This work will likely be of broad interest to evolutionary and ecological researchers, presenting a comprehensive and large-scale comparative study of the evolution of transparency on butterfly wings. The authors find that transparency has repeatedly evolved in mimicry rings, with sometimes similar underlying wing micro and nano structures. The authors suggest that wing transparency may be an aposematic, or warning signal advertising chemical defenses, in addition to camouflage.

---

## [Decision Letter]

**Decision letter after peer review:**

[Editors’ note: the authors submitted for reconsideration following the decision after peer review. What follows is the decision letter after the first round of review.]

Thank you for submitting the article "Mimicry drives convergence in structural and light transmission features of transparent wings in Lepidoptera" for consideration by *eLife*. Your article has been reviewed by three peer reviewers, including Lauren A O'Connell as the Reviewing Editor and Reviewer #1, and the evaluation has been overseen by a Senior Editor. The following individual involved in the review of your submission has agreed to reveal their identity: Nicola Nadeau (Reviewer #2).

We are very sorry to say that, after extensive consultation between the reviewers, we have decided that your article will not be considered further for publication by *eLife*. One of the reviewers raised concerns that there are serious limitations in the way the transmittance data were collected and that the conclusions of the` manuscript are not well supported. If you wish to respond to these comments, we would be willing to consider a well-argued appeal.

*Reviewer #1:*

Pinna et al. present a comparative study of transparent wings in butterflies and show that optical properties of transparent patches are convergent in co-mimics using predator vision modeling. They also examined the micro and nano structures of the transparent wings and found large structural diversity across species that influence light transmission. The data is largely appropriately analyzed and the figures display the data beautifully.

The conclusions are mostly well supported, although some aspects of the framing should be a little more careful. The authors discuss their results in the context of aposematism and mimicry, but the chemical defenses seem to be unknown and crypsis could also be important. More clarity is needed within this framework.

1. Line 63 – It seems like McClure et al. shows predator learning, but not unpalatability, as implied in this sentence. Please provide a reference that provides evidence that these transparent-winged butterflies have chemical defenses.

2. Figure 2. This is amazing. Really stunning and exciting.

3. Paragraph at 328. Does transparency correlate with chemical defenses across species? This may be known from the literature and is linked to one of the main questions of “whether transparency is involved in an aposematic signal”.

4. There were many samples collected in the tropics (Ecuador and Peru), but no local scientists are co-authors or listed in the acknowledgements. Could this have been an oversight? Were these specimens mainly from museum collections or were they captured for this project?

*Reviewer #2:*

The authors have generated an impressive data set of transparency and nanostructure measurements from 62 species of butterflies and moths belonging to 10 mimicry rings in Ecuador and Peru. They use this to assess the extent of evolutionary convergence between co-mimetic species and to determine the relationships between different micro and nano structures and the level of transparency.

Strengths:

The large numbers of species analysed, including convergence at different evolutionary distances. Detailed measurements taken from all of these species. Robust analysis methods.

Weaknesses:

It appears that only scales and structures on the ventral side of the wing were measured, but both sides will be important in determining transparency. Only wing membrane nanostructures are considered, not scale nanostructures, which could also be important for transparency.

The authors convincingly show that co-mimetic species are convergent in both the level of transparency and, to some extent, in the structures they have. This is fascinating as it suggests that the extent of transparency is something that predators use to assess profitability of prey. Previously transparency was mostly thought to be used mainly to avoid detection by predators.

The analysis of the effectiveness of the different structures in increasing transmittance of light is also very interesting, showing that pillar and sponge-like nanostructures appear to be the most effective. This is the first description of these sponge-like structures. These results could be of interest in producing effective non-reflective structures. An area which has already taken inspiration from biological systems.

Were p-values corrected for multiple testing? There are a large number of tests performed, so perhaps this should be done, or the significance threshold should be adjusted. I think the main results will hold anyway.

In the discussion, "Gomez et al. 2020 in review" is cited a few times and in places it seems that some of the new findings of this papers are also reported there (eg the sponge-like structures and transparent scales). It would be helpful to clarify some of those points. In particular, the sentence starting on line 385 is unclear "Transparent scales have already been reported in the opaque papilionid Graphium sarpedon (Stavenga et al., 2010) and as Gomez et al. (2020, in review) we are describing them for the first time in transparent Lepidoptera."

*Reviewer #3:*

Pinna et al. used spectrophotometry, comparative methods, and microscopy to investigate the convergent evolution of transparent wing patches across 10 groups of Mullerian mimics. The authors provide abundant new data on the micro- and nanostructures of transparent wing patches, potentially offering insights into necessary features of transparent wings and forming a comparative dataset to investigate tradeoffs associated with transparency. In addition, the authors claim that transparent wing patches are more similar within mimicry rings than expected based on phylogenetic distance in support of the hypothesis that transparent wing patches may form part of the aposematic signal in these Lepidopterans. While the question is an interesting one, there are major concerns about the spectrophotometric methods used to collect transmittance data and whether the existing data supports the authors' claim.

Weaknesses:

1) Transmittance data forms the foundation of most of the analyses in this paper, however the reported transmittance measurements reveal either flawed methodology or a flawed premise. Many wing patches measured here have transmittance values of <10% (including some with transmittance <1%), which would make these wing patches opaque, if the data are accurate. This issue may be caused by the spectrophotometric technique used. By measuring the transmittance of an object that is not flat with several centimeters between the object and the detector, small amounts of scattering or refraction can lead to large errors in measured transmittance. For example, a glass prism in a similar experimental set up would lead to a measurement of ~0% transmittance, despite being perfectly transparent. Without accurate transmittance data, it is difficult to evaluate the strength of the authors' claims.

2) Beyond the issues with the transmittance data, it is not clear that the authors' conclusions are well supported by the data. When investigating the convergence of transmission properties, the pooled dataset of all species seems to demonstrate convergence across 4/5 wing patches, but when broken out by mimicry ring the story is different. Only 2/10 mimicry rings show evidence of convergence in at least 5/10 comparisons (5 chromatic, 5 achromatic comparisons per group). Conversely, 4/10 mimicry rings show no evidence of convergence. The discussions of the findings within the manuscript seem to focus heavily on the pooled data and do not discuss the substantial differences in evidence for convergence across mimicry rings.

3) Similar to the investigation of convergent transmission properties, the tests for micro- and nanostructure convergence seem to show substantial differences across mimicry rings. Despite the pooled data showing convergence in scale scale type, only 2/10 mimicry rings show evidence for scale type convergence. Focused on nanostructures, 5/10 mimicry rings show evidence for nanostructure convergence, but the authors' analyses later in the manuscript demonstrate that nanostructure type has no effect on transmittance in any of the 8 retained models.

Strength:

The authors have generated an exciting comparative dataset of the micro- and nanoscale scale features that may be involved in transparency in Lepidoptera. Datasets comprised of this many species are uncommon in biophotonics. Further quantification of these structures paired with accurate transmittance data could provide a number of insights into the structural basis of terrestrial transparency, a less well understood phenomenon than aquatic transparency.

My primary recommendation for improving the manuscript center around the collection of transmittance data. To reduce the effects of scattering and refraction on the measurement, it is necessary to reduce the distance between the emitting and collecting probes to as little as possible. Over a 7cm distance, even small errors (2 degrees or less) can drastically affect the result. Another potential option is to use integrating spheres on both sides of the wing to collect transmittance regardless of whether light was scattered or not.

It would also be useful to the reader to see a discussion of the differences between the pooled results and the convergence results within each mimicry ring. There are potentially interesting reasons for why one might find convergence in some groups and not others, but those are not explored in the manuscripts current form.

---

## [Author Response]

[Editors’ note: The authors appealed the original decision. What follows is the authors’ response to the first round of review.]

Reviewer #1:Pinna et al. present a comparative study of transparent wings in butterflies and show that optical properties of transparent patches are convergent in co-mimics using predator vision modeling. They also examined the micro and nano structures of the transparent wings and found large structural diversity across species that influence light transmission. The data is largely appropriately analyzed and the figures display the data beautifully.The conclusions are mostly well supported, although some aspects of the framing should be a little more careful. The authors discuss their results in the context of aposematism and mimicry, but the chemical defenses seem to be unknown and crypsis could also be important. More clarity is needed within this framework.

Many thanks for your positive comments. We now add more information about the presence of chemical defences in clearwing species (see below).

1. Line 63 – It seems like McClure et al. shows predator learning, but not unpalatability, as implied in this sentence. Please provide a reference that provides evidence that these transparent-winged butterflies have chemical defenses.

Our wording was probably confusing: in fact, McClure et al. do show that transparent ithomiine species are highly unpalatable, due to the presence of pyrrolizidine alkaloids in their body. We add this information in brackets in lines 65-66.

2. Figure 2. This is amazing. Really stunning and exciting.

Thank you.

3. Paragraph at 328. Does transparency correlate with chemical defenses across species? This may be known from the literature and is linked to one of the main questions of “whether transparency is involved in an aposematic signal”.

The only paper that looks at the link between the degree of transparency and unpalatability is McClure et al. 2019. The authors find that all tested transparent species (all belonging to the tribe Ithomiini) are highly unpalatable, and on average more unpalatable than opaque species. We recall in lines 308-309 in the discussion that these lepidoptera are chemically defended to argue that convergence on transmission properties is likely due to predators’ pressure.

4. There were many samples collected in the tropics (Ecuador and Peru), but no local scientists are co-authors or listed in the acknowledgements. Could this have been an oversight? Were these specimens mainly from museum collections or were they captured for this project?

Thanks for pointing this out. Most specimens were collected by Marianne Elias over several years of field trips in Peru and Ecuador, and some were provided by Nipam Patel. We added to the acknowledgement section (l. 635 to 639) the local colleagues and assistants that participated to field trips and those that assisted in collecting permit applications (a crucial step for research).

Reviewer #2:The authors have generated an impressive data set of transparency and nanostructure measurements from 62 species of butterflies and moths belonging to 10 mimicry rings in Ecuador and Peru. They use this to assess the extent of evolutionary convergence between co-mimetic species and to determine the relationships between different micro and nano structures and the level of transparency.Strengths:The large numbers of species analysed, including convergence at different evolutionary distances. Detailed measurements taken from all of these species. Robust analysis methods.Weaknesses:It appears that only scales and structures on the ventral side of the wing were measured, but both sides will be important in determining transparency.

We did not include data on dorsal side because preliminary study on 12 Ithomiini species showed no difference in scale density and scale dimensions between ventral and dorsal sides. We now add this information as the supplementary result 2 in the Appendix and in Supplementary file 3f, g and h in the supplementary material.

Only wing membrane nanostructures are considered, not scale nanostructures, which could also be important for transparency.

It is true that scale nanostructures may affect transparency in conjunction with membrane nanostructures, but this only holds in the case of lamellar flat transparent scales. For pilliform scales and erected lamellar scales, most of the light goes directly through the wing membrane and transparency is mostly impacted by wing membrane nanostructures. Moreover, for all types of coloured scales, pigments in the scales absorb light and considerably reduce transmission through scales. Therefore, nanostructures of those scales likely have a limited, if any, effect on transparency.

In our dataset, only 4 out of 62 species (*Hypocrita strigifera*, *Vila azeca*, *Vila emilia*, *Stalachtis euterpe*) have lamellar flat transparent scales. Because in most species scale nanostructures have no, or very little effect compared to membrane nanostructures, we feel it is not relevant to include them in our models. In fact, including them could even lead to misleading conclusions, for instance if opaque scales harbour particular types of nanostructures that have effectively no incidence on transmittance.

The authors convincingly show that co-mimetic species are convergent in both the level of transparency and, to some extent, in the structures they have. This is fascinating as it suggests that the extent of transparency is something that predators use to assess profitability of prey. Previously transparency was mostly thought to be used mainly to avoid detection by predators.The analysis of the effectiveness of the different structures in increasing transmittance of light is also very interesting, showing that pillar and sponge-like nanostructures appear to be the most effective. This is the first description of these sponge-like structures. These results could be of interest in producing effective non-reflective structures. An area which has already taken inspiration from biological systems.

Many thanks for your encouraging words! It is true that deciphering the optical impact of this diversity of nanostructures is of high interest and we are currently investigating it in more details.

Were p-values corrected for multiple testing? There are a large number of tests performed, so perhaps this should be done, or the significance threshold should be adjusted. I think the main results will hold anyway.

Thanks for pointing this out. We now apply the ‘Holm’ method to correct p-values for each spot and each combination of illuminant and visual system and for each structural feature. Such a correction is more powerful than Bonferroni correction. In figures 1 and 4, in figure 1 —figure supplements 1 and 3 and in figure 4 —figure supplement 1 we present the corrected p-values. In supplementary files 1a and 1b we present both raw and corrected p-values.

In the discussion, “Gomez et al. 2020 in review” is cited a few times and in places it seems that some of the new findings of this papers are also reported there (eg the sponge-like structures and transparent scales). It would be helpful to clarify some of those points. In particular, the sentence starting on line 385 is unclear “Transparent scales have already been reported in the opaque papilionid Graphium sarpedon (Stavenga et al., 2010) and as Gomez et al. (2020, in review) we are describing them for the first time in transparent Lepidoptera.”

Both papers (Gomez et al. and this one) were initially submitted almost simultaneously. Both papers reveal transparent scales in clearwing Lepidoptera (Stavenga et al. 2010 did report transparent scales, but in an opaque species), but only our paper submitted to *eLife* reports sponge-like nanostructures. Since Gomez et al. is now published in Ecological Monographs, we do not claim anymore to be the first to reveal transparent scales and we modified the sentences in the discussion (lines 338-339) as well as in the introduction (lines 84-85).

Reviewer #3:Pinna et al. used spectrophotometry, comparative methods, and microscopy to investigate the convergent evolution of transparent wing patches across 10 groups of Mullerian mimics. The authors provide abundant new data on the micro- and nanostructures of transparent wing patches, potentially offering insights into necessary features of transparent wings and forming a comparative dataset to investigate tradeoffs associated with transparency. In addition, the authors claim that transparent wing patches are more similar within mimicry rings than expected based on phylogenetic distance in support of the hypothesis that transparent wing patches may form part of the aposematic signal in these Lepidopterans. While the question is an interesting one, there are major concerns about the spectrophotometric methods used to collect transmittance data and whether the existing data supports the authors' claim.Weaknesses:1) Transmittance data forms the foundation of most of the analyses in this paper, however the reported transmittance measurements reveal either flawed methodology or a flawed premise. Many wing patches measured here have transmittance values of <10% (including some with transmittance <1%), which would make these wing patches opaque, if the data are accurate. This issue may be caused by the spectrophotometric technique used. By measuring the transmittance of an object that is not flat with several centimeters between the object and the detector, small amounts of scattering or refraction can lead to large errors in measured transmittance. For example, a glass prism in a similar experimental set up would lead to a measurement of ~0% transmittance, despite being perfectly transparent. Without accurate transmittance data, it is difficult to evaluate the strength of the authors' claims.

The reviewer is concerned about the validity of our transmittance measurements, performed using fibres placed several centimetres apart. Below we provide new data and arguments for why we believe our measurements are robust. We also make clearer that patches with very low transmittance are opaque, or nearly so.

Collimated fibres and specular transmission

As stated in the previous version (Material and methods in the Appendix, l.49 and l.51), the fibres are collimated over a distance of 11 cm, meaning that the light beam is parallel along 11 cm and that light divergence is negligible over this distance. In our setting, the fibres are 22 cm apart, meaning that the distance between the fibres and the sample is 11 cm (now corrected in the ‘Spectrophotometry’ subsection of the material and methods section in the Appendix, l.53). We indeed mistakenly stated in the previous supplementary material that the fibres were 14 cm apart because 14 cm is the graduation of the position of the fibre. However, the sample is placed at a graduation of 25 cm, meaning that the distance between the fibres and the sample is 11 cm.

Regarding potential light scattering by the wing material, we recall that in this paper we are interested in specular transmission, which is the definition of transparency, and which is also more biologically relevant, since this is what predators perceive, and not total transmission. The collecting fibre only collects light transmitted in its direction, which is what we need. As the wing membrane is thinner than 2 μm, we assume that no refraction can deviate the light beam enough so as it is not collected by the collecting fibre.

Repeatability

In the previous version we showed data on repeatability, estimated on 19 species (either the same specimen measured several times and/or different specimens of the same species measured one time, see ‘Spectrophotometry’ subsection in the ‘Materials and methods’ section of the Appendix in the manuscript, l.87). For each measure, samples were taken and placed between the fibres (when the same sample was measured multiple times, it was repeatedly removed and replaced). Although we placed each sample perpendicularly to the fibres, we may expect small variations in the incident angle of the membrane with fibres if the wing is not perfectly flat. Yet, despite such variation, measurements are repeatable at different levels (species and spots), meaning that variation in orientation induced by placing the specimen, if any, does not affect the result and that the measurements are reliable.

Additional measurements to assess the robustness of our data

To further assess the robustness of our measurements with regard to distances between fibres and incident angle, we selected 8 species such as to maximize the diversity in macroscopic aspect of their transparent patches and the diversity of structures (one specimen per species):

On each specimen, we performed measurements at one point (the fourth spot as represented on figure 1 in the manuscript), at different distances (at 11, 9, 7, 5, and 4 cm, which is the shortest distance possible on our device) and at different incidence angles between the sample and the fibres (every 2° from -10° to 10°, which was visually far beyond any accidental variation in sample positioning, as for the original measurements of the study we ensured the sample was set perpendicular to the axis of fibres). Hence, for each species, we obtained a total of 55 spectra (5 distances and 11 angles for the same specimen). We extracted the mean transmittance (B2) for each spectrum, and we explored the effect of distance and incident angle on mean transmittance by applying a linear model, where mean transmittance B2 was the response variable and with species, distance, incident angle and their interaction as explanatory variables (Author response table 1, Author response table 2).

**Author response table 2. sa2table2:** Results of the linear model linking mean transmittance (B2) with species, distance and incident angle between fibres and the wing sample and the interaction between incident angle and distance (F_10, 429_ = 2691, p-value < 0.001, R^2^_adj_ = 0.984). Species are represented by their initials (GP: Godyris panthyale, IA: Ithomia agnosia, OO: Oleria onega, EN: Eresia nauplius, HE: Hyalurga egeon, IF: Ithomiola floralis, MC: Methona curvifascia, PIER1: Pieridae 1). The p-values presented are equal to the p-value of the mixed model adjusted for type III sums of squares.

Variable	Estimate(mean ± sd)	t-value	p-value	
Intercept	25.43 ± 0.38	67.0	< 2*10^-16^ ***	
Species	Piliform coloured scales>Lamellar scales(GP, IA, OO) > (EN, HE, IF, MC, P1)	3.89 ± 0.03	122.3	< 2*10^-16^ ***
	Within Lamellar scales:Lamellar erect scales > lamellar flat scales(HE, IF, MC) > (EN, P1)	3.10 ± 0.06	48.7	< 2*10^-16^ ***
	Within lamellar erect scales: transparent scales> coloured scales(IF) > (HE, MC)	3.80 ± 0.14	26.7	< 2*10^-16^ ***
	Within piliform coloured scales:pillars nanostructures > sponge nanostructures(GP, IA) > (OO)	7.94 ± 0.14	55.7	< 2*10^-16^ ***
	Within lamellar erect coloured scales, nipples nanostructures > absent nanostructure(HE) > (MC)	1.19 ± 0.25	4.8	2.1*10^-6^ ***
	Within piliform coloured scales:Ring panthyale > ring agnosia(GP) > (IA)	18.55 ± 0.25	75.2	< 2*10^-16^ ***
	Within opaque species:Ring lerida>ring eurimedia(EN) > (P1)	-1.52 ± 0.25	6.18	1.5*10^-9^ ***
Distance between fibres and sample	-0.0231 ± 0.005	-4.81	2.09*10^-6^ ***	
Incident angle between fibres and sample	0.056 ± 0.060	0.943	0.346	
Interaction between distance and incident angle	-0.0017 ± 0.00076	-2.36	0.019 *	

**Author response table 1. sa2table1:** 

Appearance	Species	Abbre-viation	Mimicry ring	Microstructure
Highly transparent	*Godyris panthyale*	GP	panthyale	piliform coloured scales
	*Ithomia agnosia*	IA	agnosia	piliform coloured scales
Slightly scattering	*Oleria onega*	OO	lerida	piliform coloured scales
	*Hyalurga egeon*	HE	lerida	lamellar erect coloured scales
Moderately transparent	*Methona curvifascia*	MC	confusa	lamellar erect coloured scales
	*Ithomiola floralis*	IF	agnosia	lamellar erected transparent scales
Opaque	Pieridae1	P1	eurimedia	lamellar flat coloured scales
	*Eresia nauplius plagiata*	EN	lerida	lamellar flat coloured scales

To illustrate the effect of distance, we plotted the range of transmittance for each species observed at the different measured distances, at an incident angle of 0 degree (Author response image 1). To illustrate the effect of angle, we plotted the range of transmittance for each species observed for the different measured angles, at a distance of 11 cm. (Author response image 2).

**Author response image 1. sa2fig1:** Effect of distance between fibres and wing sample on mean transmittance for each species (normal incidence: 0°). Each point represents the measurement of mean transmittance for an incident angle of 0 degree (which corresponds to our original setting) and a distance between fibres and sample ranging from 4 to 11 cm. A photo of the forewing against a chequered background is presented below the graph and the scale bar represent 1 cm.

**Author response image 2. sa2fig2:** Effect of incident angle between fibres and wing sample for each species considered at a distance of 11 cm (which corresponds to the original setting). Each point represents the measurement of mean transmittance at a distance of 11 cm between fibres and sample and at an incident angle ranging from -10° to 10°. A photo of the forewing against a chequered is presented below the graph and the scale bar represent 1 cm.

The results of the linear model (Author response table 2) show that most variables have an effect on mean transmittance except the incident angle between fibres and wing sample. As expected, the variable **‘**species**’** has a strong – and in fact, the strongest – effect on mean transmittance, as different species (each represented by a single specimen in our test) harbour different degrees of transparency (Author response image 1 and Author response image 2) .

Distance between fibres and wing sample also has a significant effect on mean transmittance: mean transmittance tends to slightly decrease as distance between fibres and wing sample increases. However, the effect size associated with distance between fibres and sample is much smaller than the effect size associated with species (0.02 for distance against 1.19 to 18.55 for species), meaning that species effect is at least 59 times higher than the distance effect (see also Author response image 1 to visualise differences between measurements within the same species and differences between species). Moreover, we can see on Author response image 1 that the effect of distance on mean transmittance is different from one species to another, meaning that there is no systematic underestimation or overestimation of transmittance values for a given distance. For example, for scattering species (*Hyalurga egeon* and *Oleria onega*), distance tends to increase mean transmittance while for highly to moderately transparent species (*Godyris panthyale*, *Ithomia agnosia*, *Ithomiola floralis*), distance tends to decrease mean transmittance. Contrarily to what was stated by reviewer 3, there is no extinction of transmittance with distance between fibres and there is no optimal distance for measuring wing transmittance, as long as the same distance is used for every measurement.

Based on the results of the linear model, incident angle does not have any significant global effect on mean transmittance (non-significant effect of incident angle in Author response table 2). Even though mean transmittance can vary with incident angle (Author response image 2), this variation is different from one species to another, meaning that there is no systematic trend overall. Moreover, the range of angles we explored in this new measurement session is much wider than the range of angles that was included in the actual measurements (which was due to small errors in wing positioning and wing roughness). This means that the small variation in angle in our original dataset has virtually no effect on our transmittance measurements.

Finally, the coefficient associated with the interaction between distance and angle is significantly different from zero and is negative, meaning that when the distance between fibres and sample increases the effect of angle decreases. While the analysis of the distance effect showed no optimal distance, the analysis of the incident angle effect shows that measurements gain in accuracy by placing the fibres further apart. Hence, our chosen distance was better in optimizing both the effects of distance and angle.

To conclude, based on the fact that fibres are collimated, that our measurements are repeatable, and that the effect of distance and incident angle are respectively modest and nil compared to the effect of species, we argue that the spectrophotometric method we use is reliable, reproducible and robust and that our measurements are accurate.

Low transmission values and opaque species

We had not made it very clear in the previous version of our article but some of the co-mimetic species are actually opaque or almost so, and transmittance values below 10% correspond to such opaque patches. Specifically, some of the mimicry rings (‘eurimedia’ mimicry ring for example) comprise opaque species, resembling transparent, yet coloured species from another lineage. For ‘agnosia’ mimicry ring, many species (mainly riodinids) ‘fake’ transparency using white patches and dark veins. We now make clearer in the introduction (l. 96-101) and in the material and method section (l.456-457) that some specimens are opaque. We also added supplementary figure to show the range of macroscopic transparency in our dataset (Figure 5 —figure supplement 1). Finally, we added the information in the ‘Transparency status’ column in the supplementary file 1d.

2) Beyond the issues with the transmittance data, it is not clear that the authors' conclusions are well supported by the data. When investigating the convergence of transmission properties, the pooled dataset of all species seems to demonstrate convergence across 4/5 wing patches, but when broken out by mimicry ring the story is different. Only 2/10 mimicry rings show evidence of convergence in at least 5/10 comparisons (5 chromatic, 5 achromatic comparisons per group). Conversely, 4/10 mimicry rings show no evidence of convergence. The discussions of the findings within the manuscript seem to focus heavily on the pooled data and do not discuss the substantial differences in evidence for convergence across mimicry rings.

We acknowledge that evidence for convergence only applies to some mimicry rings, patches and conditions, but we think this already provides evidence that selection can act on transparency too, and therefore that in those mimicry rings transparency is part of the aposematic signal.

Also note that our study has relatively little power, given that some mimicry rings only contain 2 (e.g. ‘blue’ mimicry ring) or 3 (e.g. ‘hewitsoni’ mimicry ring) species, and some only contain closely related species (‘hewitsoni’, ‘panthyale’ and ‘theudelinda’ mimicry rings only comprise Ithomiini species). Therefore, we may have underestimated the importance of convergence in transmittance in mimetic Lepidoptera.

Following this remark, we toned down the discussion to account for the fact that convergence is not detected in all rings (l.281-284), and we now add a statement on why some mimicry rings show convergence and others do not in the Results section (l. 161-164). We also modified our title to "Mimicry **can** drive convergence in structural and light transmission features of transparent wings in Lepidoptera".

We don’t see the fact that convergence does not apply to all patches or both types of contrast as an issue: selection is at play, even when it exerts to only some occurrences and aspects of transparency.

3) Similar to the investigation of convergent transmission properties, the tests for micro- and nanostructure convergence seem to show substantial differences across mimicry rings. Despite the pooled data showing convergence in scale scale type, only 2/10 mimicry rings show evidence for scale type convergence. Focused on nanostructures, 5/10 mimicry rings show evidence for nanostructure convergence, but the authors' analyses later in the manuscript demonstrate that nanostructure type has no effect on transmittance in any of the 8 retained models.

Similarly, as stated above, we feel that the fact that we do find convergence for micro- and nanostructures is some cases is enough to claim that selection can act on those structures. Regarding the fact that nanostructure type does not have effect on transmittance in this study (whereas nanostructure density has a strong effect), this may be caused by:

– limited statistical power due to small sample sizes for some types of nanostructures (only two species have ‘maze’ nanostructure type for example),

– the fact that nanostructure type is highly correlated to nanostructure density (phylogenetic ANOVA on nanostructure density with nanostructure type as factor: F = 26.26, p-value = 0.001, see Supplementary files 3d and e). Since nanostructure density (which is a quantitative variable therefore having the power to follow more closely quantitative variations in transmittance) comes out in the model, there is hardly any variation to explain left for nanostructure type.

The remark of reviewer #3 on nanostructure type/density led us to test for convergence on nanostructure density, which has an effect on mean transmittance, among co-mimetic species. For each pair of species, we calculated the difference in nanostructure density and we used the same test for convergence as for chromatic and achromatic distances. The results presented in Figure 4 and in Figure 4 —figure supplement 1 show that overall, nanostructure density is convergent among co-mimetic species. When looking more closely at each mimicry rings, we find convergence in nanostructure density in 3 mimicry rings out of 10, and we find that nanostructure density is more similar than expected at random for one mimicry rings which comprises closely related species.

Strength:The authors have generated an exciting comparative dataset of the micro- and nanoscale scale features that may be involved in transparency in Lepidoptera. Datasets comprised of this many species are uncommon in biophotonics. Further quantification of these structures paired with accurate transmittance data could provide a number of insights into the structural basis of terrestrial transparency, a less well understood phenomenon than aquatic transparency.

Thanks for your appreciation.

My primary recommendation for improving the manuscript center around the collection of transmittance data. To reduce the effects of scattering and refraction on the measurement, it is necessary to reduce the distance between the emitting and collecting probes to as little as possible. Over a 7cm distance, even small errors (2 degrees or less) can drastically affect the result. Another potential option is to use integrating spheres on both sides of the wing to collect transmittance regardless of whether light was scattered or not.

As stated above, we show that our spectrophotometer measurements are reliable and that distance between fibres barely affects the results. Regarding integrating spheres, we insist on the fact that we are interested in transparency, which is defined as specular transmission of light and which can be directly converted in visual impression by predators. We do not want to measure scattered light and measures with integrative spheres are therefore not appropriate for our study. Given these considerations, there is no point in changing our method to collect transmittance data.

It would also be useful to the reader to see a discussion of the differences between the pooled results and the convergence results within each mimicry ring. There are potentially interesting reasons for why one might find convergence in some groups and not others, but those are not explored in the manuscripts current form.

The overall convergence detected is indeed driven by some mimicry rings, but, as stated above, we think this already provides evidence that selection can act on components of transparency, and therefore that in those mimicry rings transparency is part of the aposematic signal. As mentioned above, we toned down the discussion and give explanations for the differences we observe between the pooled results and the convergence results within each mimicry ring. We also changed the title of the manuscript.